# A network linking scene perception and spatial memory systems in posterior cerebral cortex

Adam Steel [1][✉], Madeleine M. Billings[1], Edward H. Silson[2] & Caroline E. Robertson [1]

The neural systems supporting scene-perception and spatial-memory systems of the human brain are well-described. But how do these neural systems interact? Here, using fine-grained individual-subject fMRI, we report three cortical areas of the human brain, each lying immediately anterior to a region of the scene perception network in posterior cerebral cortex, that selectively activate when recalling familiar real-world locations. Despite their close proximity to the scene-perception areas, network analyses show that these regions constitute a distinct functional network that interfaces with spatial memory systems during naturalistic scene understanding. These "place-memory areas" offer a new framework for understanding how the brain implements memory-guided visual behaviors, including navigation.

[1] Department of Psychology and Brain Sciences, Dartmouth College, Hanover, NH, USA. [2] Psychology, School of Philosophy, Psychology, and Language Sciences, University of Edinburgh, Edinburgh EH8 9JZ, UK. [✉]email: adam.steel@dartmouth.edu

As we navigate through our world, the visual scene in front of us is seamlessly integrated with our memory of the broader spatial environment. The neural systems supporting visual scene processing in the posterior cerebral cortex[1–9] and spatial memory in the hippocampus and medial temporal lobe[10–18] are well described. But how do visual and spatio-mnemonic systems interface in the brain to give rise to memory-guided visual experience?

Two disparate lines of inquiry yield different hypotheses. On the one hand, mechanistic accounts of memory often posit that explicit recall of visual stimuli reinstates perceptual representations in visual areas[19–23], including the three areas of the scene-perception network (parahippocampal place area (PPA[2]), occipital place area (OPA; also referred to as the transverse occipital sulcus[3,24–27]), and medial place area (MPA; also referred to as retrosplenial complex[8,9,19,28–31]). However, recent studies question the co-localization of perceptual and memory-based representations and instead suggest a perception to memory transition moving anteriorly from areas of the scene-perception network[1,32–36]. These later findings are consistent with neuropsychological observations, where patients with intact perception but disrupted mental imagery, and vice versa, have been described[37–39]. Resolving this discrepancy is critical to understanding how contextual information from memory is brought to bear on visual representations in the brain. In short, do scene perception and memory share common neural substrates?

Here, we sought to describe the neural basis of perceptually- and mnemonically-driven representations of real-world scenes in the human brain. To do this, we conducted a series of experiments using fine-grained, individual-subject fMRI to assess whether these responses are co-localized in the brain (Experiment 1). Surprisingly, this experiment revealed strong evidence to the contrary: in all subjects, visually recalling real-world locations evoked activity in three previously undescribed areas in the posterior cerebral cortex, each lying immediately anterior to one of the three regions of the scene-perception network. We next characterized the functional and network properties of these "place-memory areas" (Experiments 2–3) and explicitly tested the wide-spread view that the neural substrates of perception and memory recall (i.e., mental imagery) activate shared regions of the human brain (Experiment 4). Together, our results reveal a network of brain areas that collectively bridge the scene perception and spatial memory systems of the human brain and may facilitate the integration of the local visual environment with spatial memory representations.

## Results

**Distinct topography of scene-perception and place-memory activity.** In Experiment 1, we mapped the topography of perceptual (henceforth "scene-perception") and mnemonic (henceforth "place-memory") activity related to real-world scene processing in 14 adult participants. We first independently localized the three regions of the scene-perception network[1] by comparing activation when participants viewed images of scenes, faces, and objects (Methods). Then, in the same individuals, we localized areas that showed preferential BOLD activation when participants recalled personally familiar real-world locations (e.g. their house) versus faces (e.g. their mother)[33]. We subsequently compared the topography (i.e. anatomical location and spatial extent) of the scene-perception and place-memory preferring areas.

Comparison of scene-perception and place-memory activation revealed a striking topographical pattern: in all participants, we found three clusters of place-memory activity in the posterior cerebral cortex, each paired with one of the three scene-perception regions (Fig. 1). We henceforth refer to these clusters as 'place-memory areas' for brevity. On the lateral, ventral, and medial cortical surfaces, the pairs of place-memory and scene-perception areas exhibited a systematic topographical relationship: the center-of-mass of each place-memory area was located consistently anterior to the center-of-mass of its corresponding scene-perception area in every individual participant (Lateral pairs: $t(13) = 16.41$, $p < 0.0001$, $d = 4.39$; Ventral pairs: $t(13) =$

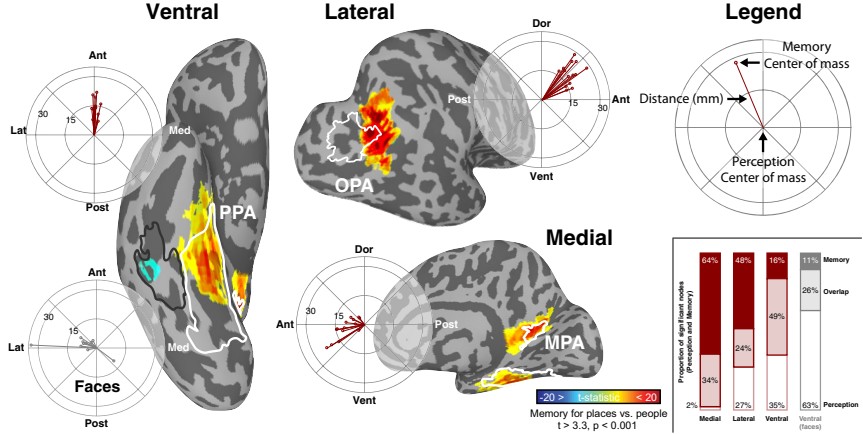

**Fig. 1 Distinct topography of place-memory and scene-perception activity in posterior cerebral cortex.** In all participants, three place-memory areas were observed, each located significantly anterior to one region of the scene-perception network. One example participant in Experiment 1 is shown (See Supplementary Fig. 1 and Supplementary Video 1 for thresholded and unthresholded activation maps for all participants ($n = 14$)). The participant's scene perception ROIs are outlined in white, and place-memory activity is shown in warm colors. The scene-perception network (parahippocampal place area [PPA], occipital place area [OPA], and medial place area [MPA]) was localized by comparing the BOLD response when participants viewed images of scenes versus with faces (outlined in white, thresholded at vertex-wise $p < 0.001$). Place-memory areas on each surface were localized in separate fMRI runs by comparing the BOLD response when participants recalled personally familiar places versus people (warm colors, thresholded at vertex-wise $p < 0.001$). Polar plots: for each cortical surface, the center of mass of place-memory activation was significantly anterior to the center of mass of scene-perception activation in all participants (all ts > 5, $p < 0.001$). In contrast, face memory activation was spatially co-localized with the face-selective fusiform face area (FFA) on the ventral surface, and no anterior shift was observed (cool colors, $t_9 = 0.1211$, $p = 0.906$). Statistical analyses revealed no difference between the hemispheres, so, for clarity, only right hemisphere is shown. Inset: while the activation during place memory was systematically anterior to activation during scene perception, the spatial overlap between perception and memory activation varied across the cortical surfaces. Note that the axes (posterior-anterior) of each polar plot are aligned to its associated cortical surface. The data in the polar plot reflects distance in millimeters.

12.115, $p < 0.0001$, $d = 3.24$; Medial pairs: $t(13) = 5.99$, $p < 0.0001$, $d = 1.6$; Supplementary Fig. 1a; Supplementary Video 1). Importantly, control analyses using (1) a more conservative contrast (scenes > faces + objects) to define scene-perception areas[40,41] and (2) a probabilistic ROI-based definition of PPA[41] confirmed that the anterior shift of memory relative to perception was not due to our definition of the scene-perception areas (Supplementary Figs. 2, 3).

Across the cortical surfaces, the degree of overlap between the scene-perception and place-memory areas varied systematically (Fig. 1, inset). On the ventral surface, place-memory activation emerged rapidly at the anterior 60–70% of the scene-perception area (PPA) and continued anteriorly beyond the scene-perception area's anterior extent (Supplementary Fig. 4). In contrast, on the medial surface, the scene-perception area (MPA) was encompassed within the most posterior portion of the place-memory responsive region. On the lateral surface, the place-memory and scene-perception area (OPA) partially overlapped, but to a lesser extent than on the other cortical surfaces. This systematic variation across the cortical surfaces was present in all participants (Supplementary Figure 1a, Supplementary Video 1).

We performed two additional analyses to characterize the cortical locations of the place-memory areas. First, we asked: do the locations of the place-memory areas fall in retinotopic, or non-retinotopic cortex? We hypothesized that, unlike scene perception areas, which process visual input, place-memory areas would lie in non-retinotopic cortex. To test this, we performed a group analysis of the place-memory localizer and compared the peak activity for each area to retinotopic maps based on a probabilistic atlas[42]. Strikingly, on all cortical surfaces, place-memory areas were consistently located anterior to the cortical retinotopic maps (Fig. 2a) with minimal overlap (Supplementary Fig. 5). This location outside of retinotopic cortex is consistent with their role as preferentially mnemonic, rather than visual, areas. Second, we asked: how do the locations of the place-memory areas relate to known anatomical landmarks in the brain? To address this question, we compared the results of the group place-memory localizer to anatomical parcels from the Glasser atlas[43]. This revealed a remarkable pattern: the peaks of the place-memory preferring activity fell at the intersection of areas known to be important for visual and spatial processing. Medially, the peak fell within the parietal occipital sulcus area 1 near the retrosplenial cortex; ventrally, the peak fell between parahippocampal area 1 and the presubiculum, and laterally the peak fell within area PGp and PGi (Fig. 2b). Thus, the place-memory areas straddle visual and spatio-mnemonic functional areas, and, based on this localization, these areas are poised to play a role in bridging these processes. In contrast, the scene-perception areas fell in retinotopic cortex and areas associated with perception (Supplementary Fig. 6).

In contrast to the striking distinction between scene-perception and place-memory, memory-related activation for faces was centered on the perceptually-driven FFA on the ventral surface (Difference of centers-of-mass: $t(9) = 0.1211$, $p = 0.906$, $d = 0.17$), and we did not observe consistent memory-driven activation near the occipital face area on the lateral surface. The anterior shift for places was greater than that for faces on all surfaces (Main effect ROI: $F(3,85) = 14.842$, $p < 0.001$, $t$ vs faces: all ps<0.001, all ds > 2.26). Importantly, face and place memories are complex and multifaceted, and factors such as vividness and memory age can influence the activity elicited by memory recall[44–46]. However, the difference between categories was not due to subjective differences in recall ability, as participants reported equal subjective vividness for place and face memory recall, although vividness of recollection was not related to activation magnitude in place-memory or scene-perception areas

(Supplementary Fig. 7). We reasoned that one potential explanation for the anterior distinction of memory versus perception for scenes/places but not faces might be the BOLD signal-dropout artifact affecting lateral ventral temporal cortex. However, we ruled out this explanation by replicating Experiment 1 using an advanced multi-echo fMRI sequence that improved signal from the lateral and anterior temporal lobe and obtaining similar results: an anterior shift for place-memory relative to scene-perception, but no distinction between the location of face memory relative to face-perception activity (Supplementary Fig. 1b). All in all, the consistent topographical relationship between perception and memory observed for scenes was not observed for faces.

These findings show that place-memory recall engages three distinct brain areas that are systematically paired with the scene-perception areas. We next sought to characterize the functional and network properties of these areas, specifically to test whether scene-perception and place-memory areas were functionally, as well as anatomically, dissociable. To ensure that we evaluated functionally homogenous regions and to control for differences in ROI sizes across regions, we constrained the scene-perception and place-memory areas to the unique members of the top 300 most scene-perception/place-memory preferring vertices for all subsequent experiments. Importantly, on all cortical surfaces, the constrained place-memory area were anterior to the scene-perception area (Supplementary Fig. 8).

**Place-memory areas respond preferentially to familiar places.** In Experiment 1, we identified three place-memory areas that showed category-selectivity for remembered places as compared with perceived scenes. But are these areas only driven by top-down, constructive memory recall tasks, or more broadly driven by memory tasks, including tasks that rely less on top-down signals, such as recognition memory? We examined this question in Experiment 2. In this experiment, participants provided a list of real-world locations that were familiar to them. Then, in the scanner, participants performed a covert recognition task, wherein they were shown panning videos of their personally familiar locations or unfamiliar locations taken from another participant's familiar locations (created using Google StreetView; Supplementary Video 2-5; see Materials and Methods). We then compared the activity of the place-memory areas with the scene-perception areas across the familiarity conditions. We hypothesized that if the place-memory areas were preferentially driven by mnemonic tasks as compared with scene areas, the place-memory network would respond more strongly than the scene-perception network when viewing videos of familiar versus unfamiliar locations.

As predicted, we found that the place-memory areas were preferentially driven by videos of familiar compared to unfamiliar locations, and, critically, this difference was significantly greater than we observed in the scene-perception areas (Region × Familiarity interaction – Lateral: $F(1,91) = 20.98$, $p < 0.001$, $t(13) = 6.40$, $p < 0.001$, $d = 2.26$; Ventral: $F(1,91) = 7.00$, $p = 0.01$, $t(13) = 4.443$, $p < 0.001$, $d = 1.55$; Medial: $F(1,91) = 7.321$, $p = 0.008$, $t(13) = 7.19$, $d = 1.05$; Fig. 3a,b). The scene-perception areas were also driven more by familiar compared to unfamiliar videos (OPA: $t(13) = 4.38$, $p = 0.0001$, $d = 1.171$; PPA: $t(13) = 4.46$, $p = 0.0001$, $d = 1.192$; MPA: $t(13) = 7.231$, $p < 0.0001$, $d = 1.932$) but to a significantly lower extent than the place-memory areas, as noted above. Importantly, the familiarity effect was not observed in control regions, the amygdala (Supplementary Fig. 9) or early visual cortex (Supplementary Fig. 10), arguing against a purely attention-related account of this effect. However, consistent with its role in recognition memory, the hippocampus

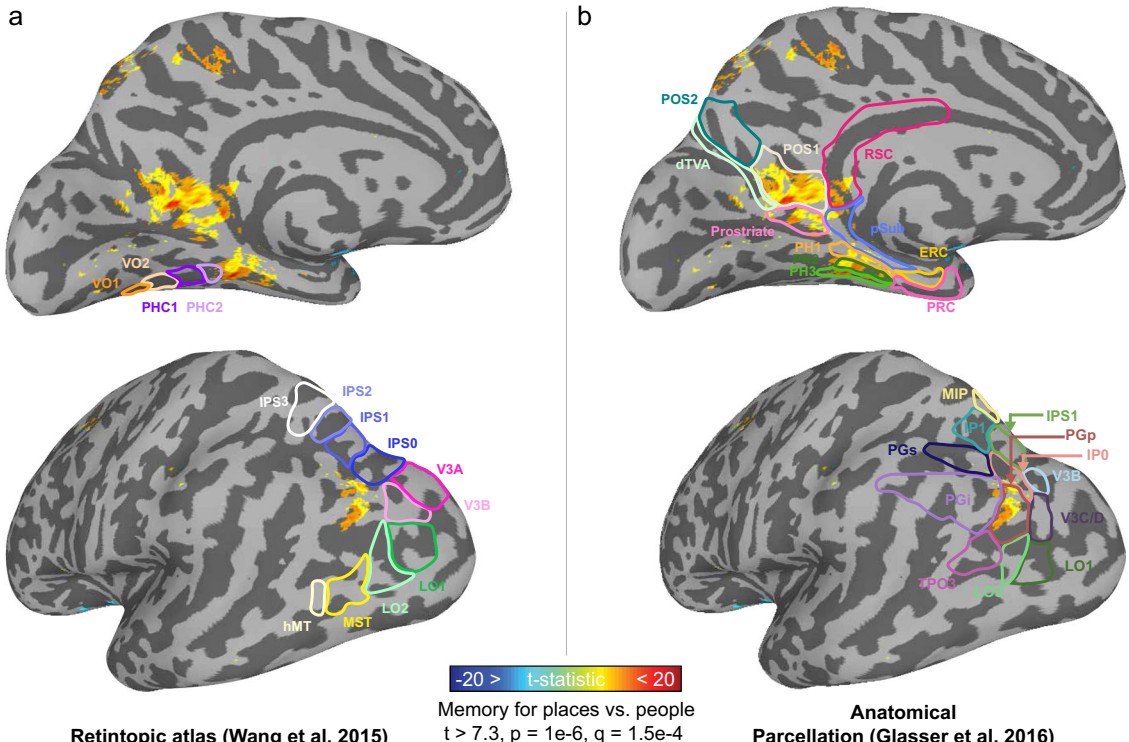

**Fig. 2 Location of place-memory activity in relation to known functional and anatomical landmarks. a** The place-memory areas are anterior to and have minimal overlap with retinotopic cortex. We conducted a group analysis of the place-memory localizer (place > people memory recall; thresholded at vertex-wise $t > 7.3$, $p = 1e-6$) and compared the resulting activation to the most probable location of the cortical retinotopic maps using the atlas (overlaid) from Wang et al. (2015)[42]. The peak of place-memory activity was anterior toretinotopic maps on the lateral and ventral surfaces and there was very little overlap between place-memory activity and retinotopic maps. **b** The place-memory areas fall at the intersection between anatomical parcels known to be involved in visual processing and those associated with spatial memory. Comparing the peaks of place-memory activity with parcels from Glasser et al. [43] (overlaid) revealed that the place-memory areas fell at the intersection of parcels associated with visual and spatial processing. Activation maps are replotted in panels a and b to allow comparison between parcellations.

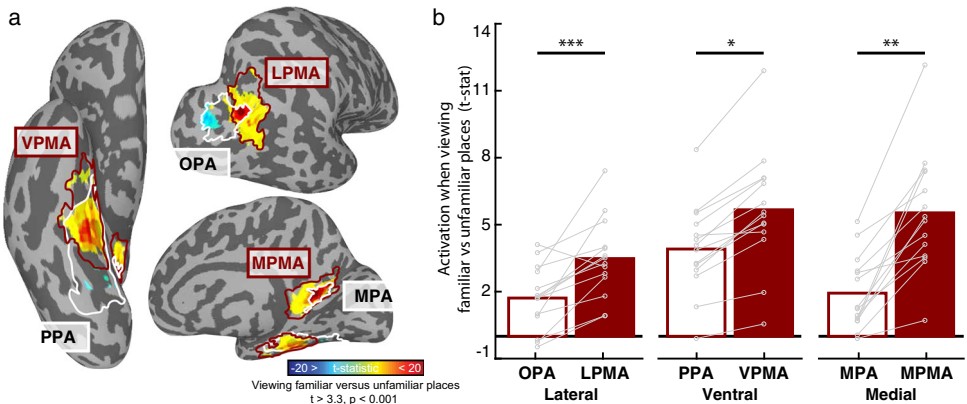

**Fig. 3 The place-memory areas respond preferentially to familiar stimuli . a**, **b** The place-memory areas preferentially activate familiar stimuli. a. Experiment 3. Participants viewed viewing panning movies of personally familiar places versus unfamiliar places, tailored to each participant using Google StreetView (see Supplementary Videos 2–5). The cortical surface depicts the contrast of BOLD activity for a single participant, thresholded at vertex-wise $p < 0.001$. Only significant vertices within the scene perception (white) and place-memory (burgundy) areas are shown. **b** Average t-statistic of vertices in the scene-perception (open bars) and place-memory areas (filled bars) when viewing videos of personally familiar places compared to unfamiliar places. On each cortical surface, the place-memory areas showed an enhanced response to familiar stimuli compared to the scene-perception areas (all ts > 2.6, ps < 0.01). Connected points depict individual participants. The hippocampus also showed a preferential response to familiar compared to unfamiliar place movies (Supplementary Fig. 9a). The amygdala (Supplementary Fig. 9b) and early visual cortex (Supplementary Fig. 10) did not show a preferential response to familiar place movies, arguing against a purely attentional account of this effect. OPA—occipital place area, LPMA—lateral place-memory area, PPA—parahippocampal place area, VPMA—ventral place-memory area, MPA—medial place area, MPMA—medial place-memory area.

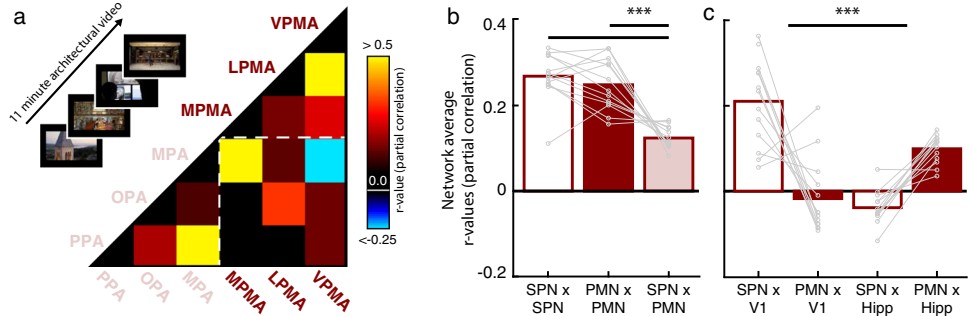

**Fig. 4 The place-memory areas constitute a distinct function network and associate closely with the hippocampus. a** Experiment 2. To assess whether the place-memory areas and scene-perception areas form distinct functional networks, participants watched an 11-minute video designed to elicit naturalistic scene understanding, comprised of several college admissions videos, real-estate listings, and architectural tours. For each participant ($n = 13$), the average time series from the scene-perception areas (pink, parahippocampal place area [PPA], occipital place area [OPA], and medial place area [MPA], the place-memory areas (burgundy, medial, ventral, and lateral place-memory areas [MPMA, LPMA, VPMA]), and the pairwise partial correlation was calculated. The correlation matrix depicts the average partial correlation of each area from all participants (ordered by Ward similarity). **b** The average pairwise partial correlation of within-network activity (Scene-perception network × Scene-perception network [SPN × SPN] and place-memory network × place-memory network [PMN × PMN]) was significantly higher than the correlation of between network activity (SPN × PMN) ($F_{(2,60)} = 50.915$, $p < 0.001$). **c** The scene-perception and place-memory areas differentially associate with the brain's visual and memory systems ($F_{(3,84)} = 55.5$, $p < 0.001$). The scene-perception areas were more correlated with early visual cortex ($t_{12} = 6.05$, $p < 0.001$), while the place-memory areas were more correlated with the hippocampus ($t_{12} = 10.64$, $p < 0.001$), which is further evidence for their roles in perception and memory, respectively. In all plots, n.s., $p > 0.05$; *$p < 0.05$; **$p < 0.01$; ***$p < 0.001$.

activated more when participants saw familiar compared to unfamiliar locations ($F(1,39) = 8.14$, $p = 0.0069$, $t(13) = 2.54$, $p = 0.004$, $d = 0.75$; Supplementary Fig. 11). These data suggest that the place-memory areas are broadly driven by memory-related task demands, including both recall and recognition memory tasks, to a greater extent than the scene-perception areas.

**Place-memory and scene-perception areas form different functional networks.** While we observed a strong dissociation between the scene-perception and place-memory areas under controlled experimental conditions, real-world behavior requires a dynamic interaction between perceptual and mnemonic processes. How do the scene-perception and place-memory areas interact under naturalistic conditions? In Experiment 3 we investigated this question. We hypothesized that, if place-memory areas and scene-perception areas were functionally distinct, these groups of areas would form separable functional networks during a naturalistic scene viewing task. Further we hypothesized that, given their functional role in place-memory tasks, place-memory areas would affiliate more with memory structures in the brain than the scene-perception areas.

Participants watched an 11-minute video that featured a concatenated series of architectural tours, college admissions videos (including Dartmouth College), and real-estate advertisements (see Methods). This video was designed to concurrently engage perceptual and mnemonic representations of complex real-world environments by engaging short-term memory built up over the course of the video, and long-term memory processes engaged during video segments that included familiar locations (i.e. Dartmouth College). We analyzed the co-fluctuation of activity among the scene-perception and place-memory areas during the movie by calculating the correlation between the activity time series for each region pair, whilst partialling out the activation of all other regions (Fig. 4a). Accounting for the activation of all other regions allowed us to assess the co-fluctuation of these areas above and beyond their shared response to the video stimulus, which causes all regions to exhibit overwhelming positively correlated activity.

This analysis revealed that the place-memory areas constitute a distinct functional network that is dissociated from the scene-

perception network: the correlation among the scene-perception and place-memory areas (i.e. within-network) was significantly greater than the correlation between these areas (i.e. between-network) (Main effect of Network: $F(60,2) = 47.99$, $p < 0.0001$; Perception:Perception v Perception:Memory, $t(12) = 7.31$, $p < 0.0001$, $d = 4.63$; Memory:Memory v Perception:Memory $t(12) = 5.31$, $p = 0.0002$, $d = 3.794$; Memory:Memory v Perception: Perception $t(12) = 1.18$, $p = 0.26$, $d = 0.48$; Fig. 4b). The difference between within- versus between-network correlations remained significant even after removing the two lowest pairwise correlation values (MPA with the ventral place-memory area and PPA with the lateral place-memory area; Perception:Perception v Perception:Memory, $t(12) = 4.88$, $p = 0.0003$, $d = 3.11$; Memory: Memory v Perception:Memory, SPN: $t(12) = 3.27$, $p = 0.001$, $d = 2.39$), which confirmed that the observed network-differentiation was not driven by outlying pairwise associations.

Next, we assessed whether the scene-perception and place-memory areas associated more with the visual and spatio-mnemonic systems, respectively, by assessng their co-fluctuations with early visual cortex and the hippocampus. Here, we observed a double dissociation (Main effect of Network: $F(3,84) = 55.5$, $p < 0.001$), whereby the scene-perception areas were more correlated to early visual cortex compared to the place-memory areas ($t(12) = 6.05$, $p < 0.001$, $d = 4.26$), while the place-memory areas were more strongly correlated with the hippocampus ($t(12) = 10.64$, $p < 0.001$, $d = 6.32$) (Fig. 4c). Together, these data demonstrate that the place-memory areas constitute a distinct functional network, which affiliates with the hippocampus during naturalistic scene understanding.

**Place-memory areas, not scene-perception areas, activate during visual mental imagery.** The observed distinction between the functional networks supporting scene perception and place memory in posterior cerebral cortex is inconsistent with a classic theory in cognitive neuroscience: that perception and recall (i.e. mental imagery) of high-level visual stimuli engage the same neural circuitry[19,29–31,44,47]. In Experiment 4, we explored this discrepancy by characterizing the relative roles of the scene-perception and place-memory areas in mental imagery and perception. Specifically, we compared activity when participants

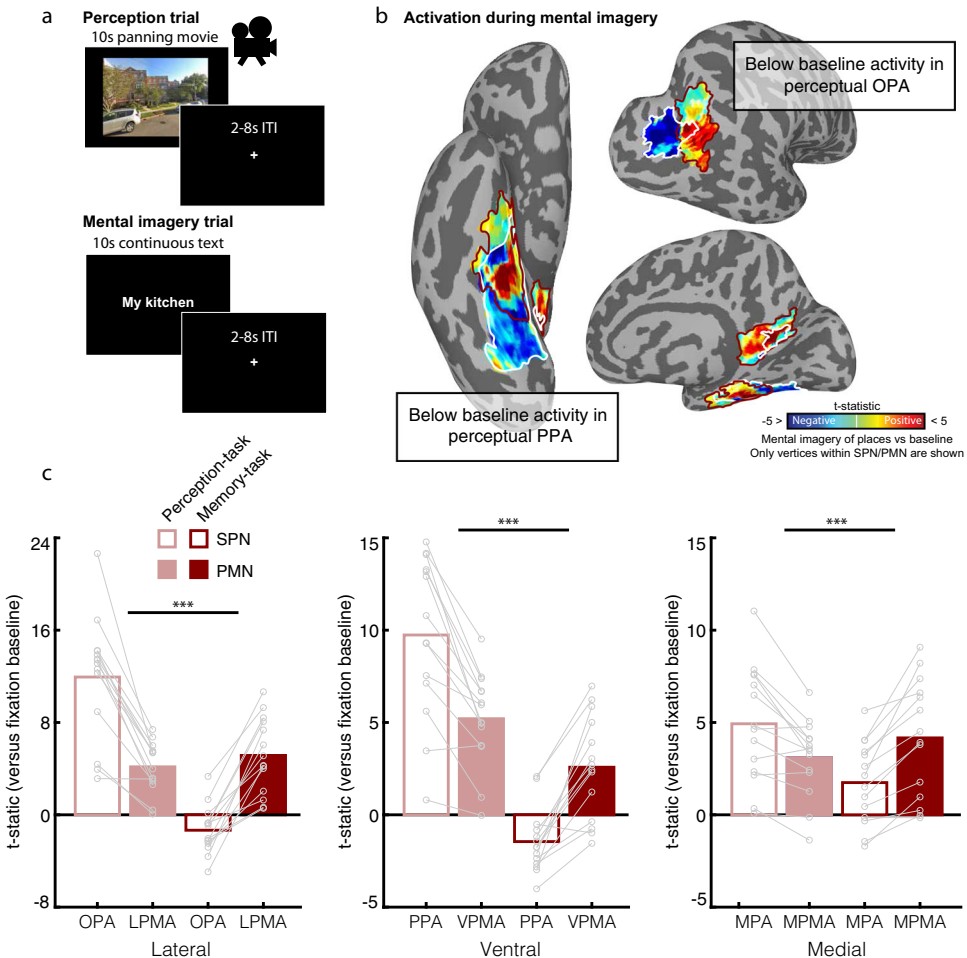

**Fig. 5 Perception and explicit memory recall (i.e. mental imagery) differentially engage the scene-perception and place-memory areas. a** In Experiment 4, participants ($n = 14$) saw panning movies of unfamiliar places (perception trials) or performed explicit memory recall (i.e. mental imagery) of personally familiar places (mental imagery trials). **b** BOLD activation during mental imagery of places compared to baseline for a representative subject. Below baseline activation within the perceptual areas PPA (ventral) and OPA (lateral) are highlighted. Only vertices within the scene-perception network (SPN; white) and place-memory network (PMN; burgundy) are shown. **c** The scene-perception areas and place-memory areas are differentially engaged during scene perception and mental imagery. Activation versus baseline of the scene-perception (open bars) and place-memory areas (filled bars) during perception of places (pink) or mental imagery of places (red). A linear mixed effects model analysis revealed that on each cortical surface, there was a significant dissociation in activation during perception and mental imagery, where the scene perception areas were significantly more active during perception, while the place-memory areas were significantly more active during mental imagery of places (ROI × Task interaction—Lateral: $F_{(1,91)} = 237.37$; $p < 0.001$; Ventral: $F_{(1,91)} = 78.19$; $p < 0.001$; Medial: $F_{(1,91)} = 28.96$; $p < 0.001$). Early visual cortex (Supplementary Fig. 11) and amygdala (Supplementary Fig. 12) also showed below baseline responses during mental imagery; hippocampus responded more to perception compared to imagery, but positively in both conditions (Supplementary Fig. 12). In all plots, n.s., $p > 0.05$; *$p < 0.05$; **$p < 0.01$; ***$p < 0.001$. OPA—occipital place area, LPMA—lateral place-memory area, PPA—parahippocampal plaace area, VPMA—ventral place-memory area, MPA—medial place area, MPMA—medial place-memory area.

performed explicit place-memory recall in a manner identical to Experiment 1 (i.e. mental imagery) to activity evoked by viewing panning videos of unfamiliar places (Fig. 5a).

Consistent with our results in Experiment 1, we found that mental imagery and perception differentially engage the place-memory and scene-perception areas, respectively (Network × Task interaction—Lateral: $F(1,91) = 237.37$; $p < 0.001$; Ventral: $F(1,91) = 78.19$; $p < 0.001$; Medial: $F(1,91) = 28.96$; $p < 0.001$). Specifically, we observed a double dissociation on all cortical surfaces. Compared to the place-memory areas, the scene-perception areas showed greater activation during perception than mental imagery (Lateral: $t(13) = 7.33$; $p < 0.001$, $d = 2.80$; Ventral: $t(13) = 6.21$, $p < 0.001$, $d = 1.80$; Medial: $t(13) = 4.15$, $p < 0.001$, $d = 0.97$; Fig. 5b, c). In contrast, compared to the scene-perception areas, the place-memory areas were more active during mental imagery (Lateral: $t(13) = 8.35$, $p < 0.001$, $d = 3.33$;

Ventral: $t(13) = 6.63$, $p < 0.001$, $d = 2.44$; Medial: $t(13) = 5.44$, $p < 0.001$, $d = 1.25$; Fig. 5b, c). Remarkably, PPA, a region that prior studies found to be active during mental imagery[29], showed below-baseline activation during mental imagery ($t(13) = -3.132$, $p = 0.008$), as did OPA ($t(13) = -2.407$, $p = 0.03$). Within the scene-perception network, only MPA showed above-baseline activation during both mental imagery ($t(13) = 2.94$, $p = 0.01$) and perception ($t(13) = 4.89$, $p = 0.003$). Early visual cortex (Supplementary Fig. 11), hippocampus (Supplementary Fig. 12a), and amygdala (Supplementary Fig. 12b) all showed greater activation during perception compared to imagery. These findings contradict the classic understanding that mental imagery recruits the same neural substrates as perception[19,29–31,44,47], and suggest that the place-memory network, not the scene-perception network, supports explicit visual recall (i.e. mental imagery) of places.

## Discussion

To summarize, we found that scene perception and place memory are subserved by two distinct, but overlapping, functional networks in posterior cerebral cortex. The three regions of the place-memory network fall immediately anterior to regions of the scene-perception network on all three cortical surfaces—PPA (ventral), MPA (medial), and OPA (lateral). Despite their topographic association with the scene-perception areas, the place-memory areas were functionally dissociable on the basis of their relative preference for mnemonic tasks: the place-memory areas responded more when participants viewed videos of familiar compared to unfamiliar places. The place-memory areas also formed a dissociable functional network during extended naturalistic scene understanding, and this network was more strongly correlated with the hippocampus than the primary visual cortex. Taken together, our data suggest that the place-memory areas might be a key interface between the scene perception and spatial memory systems in the human brain.

Multiple recent studies have suggested that anterior aspects of ventral temporal cortex, including PPA, harbor relatively more abstract[48] and context-related[35,49] information than posterior aspects, consistent with their functional connectivity with memory structures[1,32–34,50]. Our data situate these prior findings within a broader topographical organization of posterior cerebral cortex: three place-memory selective areas lie immediately anterior to the three scene-selective visual areas on all three cortical surfaces. These findings reveal a new mechanistic step between the regions of the brain that support spatial memory and those that support visual scene analysis, closing a critical gap in our understanding of how the brain integrates perceptual and memory representations.

**Topography of scene-perception and place-memory areas**. The topographical relationship between the scene-perception and place-memory areas was strikingly consistent across individuals: on all cortical surfaces, place-memory selective activity was located immediately anterior to scene-perception activity. However, intriguingly, the degree of overlap varied systematically by cortical surface. Specifically, on the ventral surface, the scene-selective area PPA was activated by scene perception and located immediately posterior to an area that was activated by both perception and memory (the ventral place-memory area, VPMA). The underlying organizational principles of ventral temporal cortex have been hotly debated. Along the medial-lateral axis, multiple organizational gradients have been identified, including eccentricity[24,26,40], animacy[51–54], and real-world size[55]. On the other hand, the anterior-posterior axis has been conceptualized as a gradient from concrete/perceptual representations to abstract/conceptual representations[40,46,56–61]. How do our results relate to these gradients? Theories of medial-lateral organization can account for the presence of scene-preferring visually responsive regions in the medial ventral temporal cortex because scene-selective PPA occurs in peripherally-biased ventral temporal cortex and responds preferentially to larger objects like houses. However, theories of anterior-posterior organization do not explain how visual scene representations in PPA are transformed into allocentric spatial representations in the anterior medial temporal lobe[11,12,16,62]. The present results address this knowledge gap, suggesting that a distinct area might subserve this process: the place-memory responsive region is well-positioned to integrate visual information represented in PPA with the allocentric spatial representations thought to reside in the hippocampus and entorhinal cortex[11,62]. Thus, by introducing areas that may interface between the visual cortex and memory structures, the present work fills this important gap in our understanding of how the brain may implement visually guided behaviors.

In contrast to the organization of the ventral surface, the scene-selective visual area on the medial surface, MPA, was activated by both scene perception and place memory and was contained within a larger place-memory responsive area (the medial place-memory area, MPMA) that extended anteriorly into the medial parietal cortex and retrosplenial cortex proper (Brodmann's area 29/30). The medial parietal cortex has traditionally been thought of as a homogenous area engaged during episodic memory processes[33,63–65]. However, many studies have focused on understanding the presence of a unique scene-preferring visual area in this region[1,5,34,66], and we and others have demonstrated that specific subdivisions of medial parietal cortex show category-preferences during memory recall, including the place-memory responsive area described here[33,63,64]. But few studies have produced an integrative account of the medial parietal cortex's mnemonic and perceptual functions because they are often investigated separately[50]. Thus, the present work unifies prior work by describing the topographical and functional distinction between MPA and MPMA: MPA may be best described as a visually responsive subregion of a larger, memory responsive area. This raises the question, what function could this organization subserve? Previous studies have found that the medial parietal cortex contains signals that reflect the local spatial environment from an egocentric perspective, including portions of the visual field outside of the current field-of-view[67]. Separately, studies in humans and animal models have shown that retrosplenial cortex proper (Brodmann area 29/30) represents the current heading direction[68–72]. Taking these results together, we suggest that that MPMA, which overlaps with the retrosplenial cortex, might represent the local spatial environment in an egocentric reference frame, with MPA reflecting a visual signal underlying this computation.

On the lateral surface, scene-selective OPA was activated by perception and located immediately posterior to an area that was activated only during memory (the lateral place-memory area, LPMA). Intersecting vertices between the two regions showed joint responsivity to both perception and memory. Among all of the surfaces, the place-memory area on the lateral surface is perhaps most topographically striking and theoretically surprising. Unlike for MPA and PPA, the relationship of OPA to memory-related processes has not been widely discussed. One study found that panoramic scene content associated with a visually presented, discrete field of view could be decoded on the lateral surface[67], suggesting a role for OPA beyond perceptual processing. However, OPA is more broadly characterized as involved in visual analysis: it has been shown to code three-dimensional information about spaces[73] as well as navigational affordances[74]. Using resting-state fMRI analyses, two studies have shown an area anterior to OPA where the BOLD response is more strongly correlated to anterior PPA compared to posterior PPA, leading to speculation that this area (sometimes referred to as caudal IPL) might be involved in scene memory, but this had not been experimentally tested[32,34]. Thus, our finding of a functionally distinct place-memory area on the lateral surface significantly advances our understanding of scene-perception and place-memory processes on the lateral surface of the human brain. Future work should focus on identifying the specific contribution of LPMA to memory-guided visual behavior.

**Content-specificity in the default mode network**. While we focused on the intersection between the place-memory and scene-perception areas, it is interesting that the place-memory network resides within the caudal inferior parietal lobule, medial temporal lobe, and posterior parietal cortex, which are also hubs of the default mode network on the lateral, ventral, and medial surfaces,

respectively[75]. Recent conceptualizations of human cortical organization consider these hubs as the apex of a macroscale gradient, spanning from unimodal (i.e. sensory or motor) areas to transmodal, high-level areas[76,77]. These transmodal areas are thought to facilitate cognitive processes, such as autobiographical memory, in a content-agnostic fashion[18,46,75,76,78,79]. Intriguingly, however, our data show that place-memory selective subregions exist within each of these default mode network hubs, broadly mirroring the category-selective subdivisions of the visual system[33,40,80]. Based on our observations here and prior work[18,33,63,64,80], we hypothesize that content-specificity may be a defining feature of cortical organization at each level of processing, from visual to transmodal cortex. Neuropsychological patients support this assertion: damage to specific areas within the caudal inferior parietal lobule, posterior parietal cortex, or medial temporal lobe cause severe navigational impairments, while leaving other memory domains, such as face memory, intact[81,82]. This content-specific organization likely persists in prefrontal areas[83–85], although traditional functional imaging analysis approaches might fail to detect tightly interdigitated features due to topographic heterogeneity across individuals[83–85]. Revealing the relationship between transmodal cortex and content-specific networks within individual participants will be paramount to understanding the neuroanatomical substrates underlying complex cognitive processes, and future studies directly comparing these systems could help to elucidate this issue.

**A bridge between perceptual and memory structures**. In addition to residing at the intersection of transmodal and perceptual cortex, the place-memory areas lie at the intersection between perceptual areas and memory structures in the brain, including the hippocampus and enthorinal cortex. Although traditionally the hippocampus has been associated with episodic memory[45,86], recently the hippocampus has been implicated in both perceptual and mnemonic tasks, including scene construction[87–89]. Consistent with this view, we found that the hippocampus, like the place-memory areas, responded preferentially to perceiving familiar stimuli compared to unfamiliar stimuli, suggesting that it has a role in visual recognition-related processes. Interestingly, unlike the place-memory areas, when we directly compared perception and recall in Experiment 4, the hippocampus responded more during perception compared to recall, which is qualitatively more similar to the scene-perception areas than the place-memory areas. Together, these experiments suggest that the hippocampus is involved in both perceptual and memory processes, but we cannot delineate what the hippocampus specifically contributes in each case. For example, during perception of unfamiliar stimuli, the hippocampus may be encoding relevant perceptual features, or comparing the present visual stimuli versus other internal representations, or both. Future work is necessary to disentangle the role of the hippocampus in scene perception and situate it in the context of the place-memory and scene-perception networks.

**Different neural substrates for perception and recall (i.e. mental imagery)**. Based on previous fMRI studies, it has been widely assumed that perception and recall (i.e. mental imagery) of high-level stimuli (such as scenes) recruit the same neural substrates, including category-selective areas in ventral temporal cortex[31,90]. For example, prior studies found that both perceiving and recalling scenes evoked activity in overlapping, scene-selective cortical areas and concluded that perception and memory shared neural substrates[22,29,47]. However, in contrast, neuropsychological studies have suggested that mental imagery

and visual recognition may have dissociable neural substrates[37,39,91], given observations of patients with preserved visual recognition but impoverished mental imagery and vice versa. Our findings reconcile this discrepancy, by showing that distinct, but neighboring regions underlie perception and recall (i.e. mental imagery) of places. These neighboring regions, which are evident at the individual-subject level, likely account for recent group-level findings of an anterior shift in category-level decoding of scenes during recall vs. perception[36]. In sum, we observe minimal overlap between the neural substrates of scene perception and place memory: the (anterior) place-memory network, but not the (posterior) scene-perception network, preferentially activates during recall (i.e. mental imagery).

So, why might previous fMRI studies have reported that the substrates of perception and imagery in the ventral temporal cortex are shared? First, the initial fMRI investigation into mental imagery and perception of high-level visual stimuli[29] used familiar exemplars for perception, which could recruit the VPMA (similar to Experiment 2 here). Second, this initial study, which used a combination of individual-subject mapping and ROI analyses, reported that only a portion of PPA was active during imagery[29], but later studies did not investigate this distinction. So, the mental imagery-related activation and decoding observed in later studies (e.g.[30]) may also have arisen from nearby memory areas. Consistent with this hypothesis, we note that if we analyze PPA and VPMA combined as a single area, we find positive activation during mental imagery in the majority of our participants (Supplementary Fig. 13). Third, typical whole-brain group analyses could obscure the topographical relationship between perception- and memory-related activity by averaging across individuals[19,31,44]. As such, our individualized analysis approach is uniquely suited to detect this fine-grained distinction. Importantly, as in some previous studies of mental imagery[47], participants in our study recalled *familiar* locations in the recall condition but viewed unfamiliar locations in the perception condition. Future work should consider mapping the evolution of place-memory area response longitudinally as stimulus familiarity develops to better understand how familiarity shapes the activity of the place-memory areas.

**Are scenes special? Distinct perceptual-mnemonic topography for scenes, not faces**. Finally, the distinct topography of the scene-perception and place-memory networks in the posterior cerebral cortex was unique to the spatial domain—we found no separable networks for face-perception and memory (although, consistent with prior work[33], we did observe a people-memory preferring area in medial parietal cortex in most participants). These findings challenge the hypothesis that a concrete/abstract posterior-anterior gradient may broadly characterize high-level visual cortex[36,58,61]. Why might a 'place-memory network', anterior to the scene-perception network, exist, while an analogous 'face-memory network' does not? We hypothesize that this distinction might relate to the 'goal' of scene processing: integrating the current field-of-view with the larger spatial context[35,67]. In contrast, the 'goal' of visual face processing is face recognition, which requires nuanced discrimination of a discrete stimulus, but not a broader visuospatial context[6,92,93]. Neuropsychological patients support this differentiation. For example, lesions of the scene-perception network can cause impairments of spatial processing while leaving scene recognition and certain navigational abilities intact[82]. On the other hand, lesions of critical face-perception areas including the occipital face area or FFA tend to produce significant and selective impairments in conscious face recognition and imagery[94,95]. In sum, we suggest that visual scene processing regions may have unique processing

requirements relative to other visual categories (e.g., object or face processing), and generalizing computational principles from other categories may be limited. To test this hypothesis, future studies should characterize the topography of mnemonic and perceptual signals in high-level visual cortex using multiple categories beyond those used here (e.g. objects, bodies, words).

**Summary**. To conclude, we have described the place-memory network: a set of brain areas in the posterior cerebral cortex that supports the recall of familiar locations. The discovery of these functionally distinct areas, residing between the brain's perception and memory systems, provides a new framework for considering the neural substrates of memory-guided visual behaviors.

## Methods

**Participants**. Fourteen adults (9 females; age = 25.7 ± 3.6 STD years old) participated in Experiments 1, 2, and 4. Of these, 13 participants took part in Experiment 3 (one participant requested to abort the scan; 8 females, age = 25.9 ± 3.6 STD years old). In addition, a subset of the original participants (N = 6; 2 females, age = 26.3 ± 4.4 STD years old) completed the multi-echo replication of Experiment 1. Participants had normal or correct-to-normal vision, were not colorblind, and were free from neurological or psychiatric conditions. Written consent was obtained from all participants in accordance with the Declaration of Helsinki and with a protocol approved by the Dartmouth College Institutional Review Board (Protocol #31288). We confirmed that participants were able to perform mental imagery by assessing performance Vividness of Visual Imagery Questionnaire[96]. All participants scored above 9 with eyes open, indicating satisfactory performance (mean = 15.2, range = 9.75-19.25).

**Procedure**. All participants took part in Experiments 1-4. In Experiment 1, we compared i) the topography of activation during a scene/face perceptual localizer with ii) the topography of place/people memory-recall related activation collected in separate fMRI runs. In Experiment 2, we characterized the differential functional responses of the place-memory and scene-perception areas during a recognition memory task, by comparing activation when participants viewed panning movies of personally familiar as compared with unfamiliar places. In Experiment 3, we tested whether the scene-perception and place-memory areas dissociated into distinct functional networks under naturalistic conditions by evaluating the correlated activity of these areas during a movie-watching task. Finally, in Experiment 4, we evaluated the relative involvement of the scene-perception and place-memory areas in perception and visual recall (i.e. mental imagery) by comparing their activation when participants performed mental imagery of personally familiar places as compared with viewing panning videos of unfamiliar places. All experiments were conducted using Psychopy3 (version 3.2.3).

### Stimuli—Experiments 1-4
*Personally familiar place- and face-memory stimulus lists*. For Experiments 1 and 4, we asked participants to generate a list of 36 familiar people and places (72 stimuli total). These stimuli were generated prior to scanning based on the following instructions:

"For your scan, you will be asked to visualize people and places that are personally familiar to you. So, we need you to provide these lists for us. For personally familiar people, please choose people that you know personally (no celebrities) that you can visualize in great detail. You do not need to be in contact with these people now—just as long as you knew them personally and remember what they look like. So, you could choose your childhood friend even if you are no longer in touch with this person. Likewise, for personally familiar places, please list places that you have been to and can imagine in great detail. You should choose places that are personally relevant to you, so you should avoid choosing places that you have only been one time, and you should not choose famous places where you have never been. You can choose places that span your whole life, so you could do your current kitchen as well as the kitchen from your childhood home."

*Familiar panning movie stimuli*. For Experiment 2, we generated panning movies depicting places that were personally familiar to our participants. To do this, participants used the GoogleMaps plug-in iStreetView (istreetview.com) to locate and download 24 photospheres of personally familiar places based on the following instructions:

"In another part of your scan, you will see videos of places that will be familiar or unfamiliar to you. You will use iStreetView to locate places with which you are familiar. iStreetView has access to all photospheres that Google has access to, as well as all of the photospheres from Google Street View. So, you should do your best to find photospheres taken from perspectives where you have actually stood in your life. Like before, please find places that are personally familiar to you (such as in front of your childhood school), rather than famous places (like the Eiffel tower), even if you have been to that famous place in your life."

Participants were also asked to specify whether a particular viewpoint would be most recognizable to them so that this field-of-view could be included in the panning movie. Once the photospheres were chosen, panning movies were generated by concatenating a series of 90° fields-of-view images taken from the photospheres that panned 120° to the right (in steps of 0.4°), then back to the left. The videos always began and ended at the same point of view, and the duration of the video was 10 s. Supplementary Videos 2-5 show examples of these videos.

*Naturalistic movie stimulus*. In Experiment 3, participants watched an 11-minute movie depicting various, real-world scenes. The movie consisted of an architectural tour, three college admissions videos, one tourism informational video, and one real-estate listing downloaded from YouTube. Specifically, the segments were: 1) an architectural tour of Moosilauke Ravine Lodge in Warren, NH (0:00-2:00), 2) an admission video of Princeton University in Princeton, NJ (2:00-3:00), 3) an admission video of Dartmouth College in Hanover, NH (3:00-3:51), 4) an admission video of Cornell University in Ithaca, NY (3:52-6:40), 5) a tourism informational video of the University of Oxford in Oxford, United Kingdom (6:40-8:53), 6) and a real-estate listing of a lake front property in Enfield, NH (8:53-11:00). Each segment included footage of people and places, ground-level and aerial footage, and indoor and outdoor footage. Videos were chosen to match closely for content, footage type, and video quality, while varying in the degree to which participants would be familiar with the places. No audio was included in the video. Participants were not asked whether places were familiar to them, although all participants were Dartmouth students/staff and therefore familiar with the Dartmouth Campus. The full video can be found here: https://bit.ly/2Arl3I0.

**Questionnaire—vividness ratings**. To ensure that participants could reliably visualize the familiar stimuli used in Experiments 1 and 4, and to confirm that there were no differences in visualization between stimulus categories, we had participants rate their ability to richly visualize each stimulus. Specifically, after the scanning session we gave the participants the following instructions:

Please rate each stimulus on how well you are able to visualize each stimulus from 1-5 (1—I cannot visualize this person/place, 5—as though I am seeing the person/place). Please rate the stimuli base upon how well you can visualize the stimulus, and not try to remember how well you visualized the stimulus during the scan.

**Procedure—Experiment 1: perceptual category localizer**. Face- and scene-selective perceptual areas were localized using a block-design functional localizer. Participants were shown images of faces, scenes, and objects (image presentation duration: 500 ms; ISI: 500 ms; block duration: 24 s; number of blocks per condition: 6). Blocks were presented continuously with no baseline periods between blocks. Participants were instructed to passively view the images and performed no task during the perceptual localizer experiment. Two runs of the localizer were performed.

**Procedure—Experiment 1: memory activation localizer**. People and place selective memory activation was localized using a procedure adapted from[33]. On each trial, participants were shown the written name of one of 36 personally familiar people and 36 places that they provided to us prior to the MRI scan (see "Personally familiar place- and face-memory stimulus lists" above). Each stimulus was presented one time during the memory localizer task. Participants were instructed to visualize this stimulus from memory as vividly as possible for the duration of the trial (10 s) following the instructions detailed below. Trials were separated by a variable inter-trial interval (4–8 s). The task was performed over 4 runs, with 9 people and place stimuli presented during each run in a pseudo-randomized order. No more than three instances of a stimulus category (i.e. person or place) could appear consecutively.

Localizer task instructions were as follows:
"In this task, you will see the name of a person or place from the list you provided us. When the name appears, I want you to perform memory recall of that stimulus. The recall should be as vivid as possible, and you should try to do this the whole time that the name is on the screen.

If the name you see is a person, I want you to imagine that person's face in as much detail as possible. You should picture their head in front of you as the person turns their head from left to right and up and down, so that you can see all of the sides of their face. They can make different expressions, as well.

The most important thing is that you are picturing their face; when some people perform memory recall, they will imagine interacting with that person or touching and hugging them. We do not want you to do this, just imagine the face. Also, do not picture their bodies, arms, or legs. Try to focus just on their face as it moves in your mind's eye.

If the name you see is a place, I want you to imagine that you are in the place. You should direct your attention around the place, looking to the left and right of you, and looking up and down, and behind you. We want you to have the experience of being in this place as much as possible.

When you are imagining places, you should imagine yourself standing in one single spot. Don't walk around the place or imagine yourself from above. Also, we want it to be as though you are in the place, rather than thinking about the objects in the place. So, as you "look" around, don't think about the individual objects or

list them in your mind. Instead, we want you to image that the place has materialized all around you, and all of the things that would be within your field of view are "visible" at the same time. For example, if you are imagining your kitchen, do not imagine the chairs, table, and windows sequentially. Instead, try to imagine all of these features at once, and that you're looking around and experiencing them as you would if you were in the room."

Between each trial, participants were instructed to clear their mind, relax, and wait for the next trial to begin.

**Procedure—Experiment 2: familiar/unfamiliar places experiment.** The goal of Experiment 2 was to determine whether the place-memory areas respond preferentially to familiar stimuli (i.e., stimuli that could be recognized) presented visually. To this end, we compared brain activation when participants saw panning videos of personally familiar places with activation when viewing panning videos of unfamiliar places. Videos were taken from Google Maps as described above (see section Familiar panning movie stimuli). To control for low-level variation in stimuli, unfamiliar videos were randomly sampled from other participants' familiar videos. Videos of Dartmouth College and Hanover, NH were excluded from the possible unfamiliar places.

The experiment was performed across 6 runs, with each run featuring 8 familiar and 8 unfamiliar place videos. There were 24 videos in each condition, and each video was seen twice. Before the experiment, participants were told that they would be seeing videos of familiar and unfamiliar places panning from left to right, then back again and that they should imagine that they were standing in that place, turning their head like the video. The instructions provided to the participants are reproduced below.

"You will see panning movies of places that you are familiar with, as well as places that will not be familiar to you. Regardless of which you see, we want you to imagine that you are in that location and the video is your perspective as you turn your head from left to right, and back again. You should not actually turn your head.

Be sure that for all scenes you are pretending you are in that location, whether or not they are familiar to you. Of course, if it is a familiar location, you might be able to predict what is coming into view. That is fine, just do your best to imagine you're in that place for both conditions."

Each trial lasted 10 s. Trials were separated by a variable inter-trial interval (4–8 s). All videos were repeated two times in the experiment: once in the first three runs, and once in the last three runs.

**Procedure—Experiment 3: naturalistic movie-watching experiment.** The goal of Experiment 3 was to characterize the network properties of the scene-perception and place-memory areas during naturalistic conditions. 13 participants took part in a single run of an 11-minute movie-watching experiment (see "Naturalistic movie stimulus" above for stimulus details). We chose a natural movie-watching task, rather than a simple resting-state scan, to avoid the possibility that mind-wandering or mental imagery might cause a decoupling of the perception and memory networks artificially[78,97].

**Procedure—Experiment 4: mental imagery versus perception experiment.** Experiment 4 explicitly tested whether the place-memory network was activated to a greater extent than the scene perception network during mental imagery. Each imaging run featured two types of trials, perception and mental imagery trials. On perception trials, participants saw a panning movie of an unfamiliar place, and were instructed to imagine that they were in that place turning their head (as in Experiment 3). On mental imagery trials, participants saw the name of a personally familiar place, and performed mental imagery of that place following the instructions given in Experiment 1 (instructed to imagine that they were in that place turning their head). This task was presented over 6 runs. Each run featured 8 familiar place word stimuli for mental imagery and 8 unfamiliar place videos. As in Experiment 3, 24 unique stimuli were used in each condition (24 place word stimuli taken from the list generated in Experiment 1, and 24 panning videos of unfamiliar places). Trials lasted 10 s and were separated by a variable inter-trial interval (4-8 s). All stimuli were repeated two times in the experiment, once in the first three runs, and once in the last three runs. Familiar stimuli for mental imagery were randomly sampled from each participant's stimulus list used during Experiment 1. Unfamiliar places videos for the perception condition were not repeated from Experiment 2.

**MRI acquisition and preprocessing.** All data were collected on a Siemens Prisma 3 T MRI scanner equipped with a 32-channel head coil at Dartmouth College. Images were transformed from dicom to nifti files using dcm2niix (v1.0. 20190902)[98], which applies slice time correction by default.

*T1 image.* For registration purposes, a high-resolution T1-weighted magnetization-prepared rapid acquisition gradient echo (MPRAGE) imaging sequence was acquired (TR = 2300 ms, TE = 2.32 ms, inversion time = 933 ms, Flip angle = 8°, FOV = 256 × 256 mm, slices = 255, voxel size = 1 ×1 × 1 mm). T1 images segmented and surfaces were generated using Freesurfer (version 6.0)[99–101].

**Single-echo fMRI**
*Acquisition.* In Experiments 1-4, single-echo T2*-weighted echo-planar images covering the temporal, parietal, and frontal cortices were acquired using the following parameters: TR = 2000 ms, TE = 32 ms, GRAPPA = 2, Flip angle = 75°, FOV = 240 × 240 mm, Matrix size = 80 × 80, slices = 34, voxel size = 3 × 3 × 3 mm. To minimize dropout caused by the ear canals, slices were oriented parallel to temporal lobe[102]. The initial two frames were discarded by the scanner to achieve steady state.

*Preprocessing. Task fMRI (Experiments 1, 3, and 4).* Task fMRI data were preprocessed using AFNI (version 20.3.02 'Vespasian')[103]. In addition to the frames discarded by the fMRI scanner during acquisition, the initial two frames were discarded to allow T1 signal to achieve steady state. Signal outliers were attenuated (3dDespike). Motion correction was applied, and parameters were stored for use as nuisance regressors (3dvolreg). Data were then iteratively smoothed to achieve a uniform smoothness of 5 mm FWHM (3dBlurToFWHM).

*Naturalistic movie watching (Experiment 2).* Naturalistic movie-watching data were preprocessed using a combination of AFNI and FSL[104] (version 6.01) tools. Signal outliers were attenuated (3dDespike). Motion correction was applied. Data were then iteratively smoothed to achieve a uniform smoothness of 5 mm FWHM (3dBlurToFWHM).

For denoising, independent component analysis (ICA) was applied to decompose the data into signals and sources using FSL's melodic[104–106]. These were classified as signal or noise by one investigator (AS) using the approach described in Griffanti et al.[107]. Components classified as noise were then regressed out of the data (fsl_regfilt). Motion from volume registration was not included in the regression, as motion is generally well captured by the ICA decomposition[107]. Time series were then transferred to the high-density SUMA standard mesh (std.141) using @SUMA_Make_Spec_FS and @Suma_AlignToExperiment.

**Multi-echo fMRI**
*Acquisition.* To better characterize activation during mental imagery of personally familiar people, we acquired a replication dataset of Experiment 1 using a multi-echo T2*-weighted sequence. Multi-echo imaging afforded the benefit of mitigating dropout with short echo times, while maintaining a high level of BOLD contrast with long echo times[108–110]. The sequence parameters were: TR = 2000 ms, TEs = [11.00, 25.33, 39.66, 53.99 ms], GRAPPA = 2, Flip angle = 75, FOV = 240 × 240 mm, Matrix size = 80 × 80, slices = 40, Multi-band factor = 2, voxel size = 3 × 3 × 3 mm. As with single-echo acquisition, the slices were oriented parallel to the temporal lobe. The initial two frames were discarded by the scanner.

*Preprocessing.* Multi-echo data preprocessing was implemented based on the multi-echo preprocessing pipeline from afni_proc.py. Initial preprocessing steps were carried out on each echo separately. Signal outliers were attenuated (3dDespike). Motion correction parameters were estimated from the second echo (3dVolreg); these alignment parameters were then applied to all echoes.

Data were then denoised using multi-echo ICA (tedana.py[109–111]). The optimal combination of the four echoes was calculated, and the echoes were combined to form a single, optimally weighted time series (T2smap.py). Data were then subjected to PCA, and thermal noise was removed using the Kundu decision tree, which selectively eliminates components that explain a small amount of variance and do not have a TE-dependent signal decay across echoes. Subsequently, ICA was performed to separate the time series into independent spatial components and their associated signals. These components were classified as signal and noise based on known properties of the T2* signal decay of BOLD versus noise. The retained components were then recombined to construct the optimally combined, denoised time series. Following denoising, images were blurred with a 5 mm Gaussian kernel (3dBlurInMask) and normalized to percent signal change.

**fMRI analysis**
*Region of interest definitions (Scene-perception and Place-memory areas, Experiment 1).* To define category-selective perceptual areas, the scene perception localizer was modeled by fitting gamma function of the trial duration with a square wave for each condition (Scenes, Faces, and Objects) using 3dDeconvolve. Estimated motion parameters were included as additional regressors of no-interest. To compensate for slow signal drift, 4th order polynomials were included for single-echo data. Polynomial regressors were not included multi-echo data analysis. Scene and face areas were drawn based on a general linear test comparing the coefficients of the GLM during scene versus face blocks as well as scene versus face + objects blocks. These contrast maps were then transferred to the SUMA standard mesh (std.141) using @SUMA_Make_Spec_FS and @Suma_AlignToExperiment. A vertex-wise significance of p < 0.001 along with expected anatomical locations was used to define the regions of interest – in particular, visually responsive PPA was not permitted to extend posteriorly beyond the fusiform gyrus.

To define category-selective memory areas, the familiar people/places memory data was modeled by fitting a gamma function of the trial duration for trials of each condition (people and places) using 3dDeconvolve. Estimated motion parameters were included as additional regressors of no-interest. To compensate for slow signal drift, 4th order polynomials were included for single-echo data. Polynomial

regressors were not included multi-echo data analysis. Activation maps were then transferred to the suma standard mesh (std.141) using @SUMA_Make_Spec_FS and @Suma_AlignToExperiment. People and place-memory areas were drawn based on a general linear test comparing coefficients of the GLM for people and place memory. A vertex-wise significance threshold of p < 0.001 was used to draw ROIs.

To control for possible signal-to-noise differences introduced by variably sized ROIs, for analysis of correlated activity during movie watching (Experiment 3) and activation in Experiments 2 and 4, scene-perception and place-memory ROIs were constrained to unique members of the top 300 vertices from the perception and memory areas. These ROIs are referred to as "constrained ROIs" in subsequent sections.

Importantly, our findings in Experiments 1-4 remained significant when two additional approaches to defining scene-selective areas were used: 1) the contrast scenes > faces + objects, and 2) a disk of 300 nodes drawn around the center of each ROI (See Supplementary Figs. 14,15).

*Analysis of topography of perception and memory areas (Experiment 1).* The topography of the perception and memory areas was compared in two ways: by identifying whether there was a significant anterior shift from perception to memory and quantifying the overlap between category-selective perception and memory vertices at the significance threshold $p < 0.001$.

To quantify the anterior displacement of perception and memory areas, we calculated the weighted center of mass for the scene-perception/place-memory selective area on each surface, where the contribution of each vertex to the center of mass was weighted by its selectivity (*t*-statistic). The distance between the center of mass for perception and memory in the *y*-dimension (posterior-anterior) was then compared using a linear mixed effects model with ROI (perception/memory) and Hemisphere (lh/rh) as factors separately for each surface. Because there was no significant effect of hemisphere, the data are presented collapsed across hemisphere (all ps>0.10). In addition, to determine whether the anterior displacement of memory compared to perception was specific to scenes, we compared the distance in the y-direction between the PPA and ventral place-memory area (VPMA) with the FFA and the face-memory selectivity are on the ventral surface using a linear mixed effects model with Category (scene/face), ROI (perception/memory) and Hemisphere (lh/rh) as factors. Effect sizes (Cohen's D) were calculated for all pairwise comparisons.

To further examine the relationship between perception and memory on each surface, we then qualitatively compared the overlap between the category-selective perception and memory areas on each surface.

*Group place-memory localizer analysis.* Group place-memory activity was defined by comparing beta-values during place- versus people- memory recall activation for each subject using 3dttest++ in AFNI. Data were thresholded at vertex-wise $q <$ 0.00015. The peak of the group localizer results was compared with (1) a prob-abilistic retinotopic atlas[42] and (2) an anatomical parcellation based on multi-modal data from the Human Connectome Project[43] aligned to the SUMA standard mesh (std.141).

*Anatomical region of interest definitions (hippocampus and primary visual cortex).* Hippocampus and primary visual cortex (occipital pole) were defined for each participant using FreeSurfer's automated segmentation (aparc + aseg).

*Task fMRI analyses (Experiments 2 and 4).* For Experiments 2 and 4, a gamma function of duration 10 s was used to model responses for trials in each condition (Experiment 2: familiar/unfamiliar places videos; Experiment 4: mental imagery/ perception). These regressors, along with motion parameters and 4th order poly-nomials were fit using 3dDeconvolve. Activation maps were then transferred to the SUMA standard mesh (std.141) using @SUMA_Make_Spec_FS and @SUMA_AlignToExperiment.

For analysis, the average t-statistic (compared to baseline) for each condition (Experiment 3: familiar versus unfamiliar videos; Experiment 4: mental imagery versus perception tasks) from the constrained ROIs was calculated. The average t-statistics were compared using a linear mixed effects model[112] in R[113] for each surface separately (i.e. PPA v VPMA, MPA v MPMA, OPA v LPMA were separately compared). Trial condition (Experiment 3—Familiarity: familiar/ unfamiliar; Experiment 4—Task: perception/imagery) and Hemisphere (lh/rh) were included as factors. There was no significant effect of hemisphere in any test, and thus data are presented collapsed across hemisphere (all ps>0.10). Post-hoc tests were implemented using the emmeans package[114]. In addition, to determine whether a region was significantly active above or below baseline, the t-static from each ROI was compared versus zero. T-statistics were chosen to aid comparison across areas, which is typical in these studies[6]. Effect sizes (Cohen's D) were calculated for all pairwise comparisons.

*Analysis of correlated activity time series during movie watching (Experiment 3).* In Experiment 3, we analyzed the co-fluctuation of activity patterns in each ROI during a naturalistic movie-watching task. For each participant, we first extracted the average time course of each constrained ROI, as well as the hippocampus and early visual cortex, which were anatomically defined based on each participant's

FreeSurfer segmentation/parcellation. We then calculated the correlation of the time series from each region pair while partialing out the time series from all other region pairs. The correlation matrices were calculated for each hemisphere sepa-rately. The average pairwise correlation within- and between- networks (percep-tion: PPA, MPA, OPA; memory: VPMA, medial place-memory area (MPMA), and lateral place-memory area (LPMA)) were then compared using a linear mixed effects model with Connection (P×P, M×M, and P×M) and Hemisphere (lh/rh) as factors. The average correlation of each network with hippocampus and early visual cortex was compared using a linear mixed effects model[112] with Network (Per-ception/Memory) and Hemisphere (lh/rh) as factors. Significant model terms were determined using an analysis of variance implemented in R[113]. There was no significant effect of hemisphere in any test, and thus data are presented collapsed across hemisphere (all ps>0.10). Effect sizes (Cohen's D) were calculated for all pairwise comparisons.

**Reporting summary**. Further information on research design is available in the Nature Research Reporting Summary linked to this article.

## Data availability
The data that support the claims of this study are available upon reasonable request to the corresponding author.

## Code availability
The code used in this study is available upon reasonable request to the corresponding author.

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

## Acknowledgements

The authors thank Ratan Murty, Iris Groen, Martin Hebart, and Brad Duchaine for comments on drafts of this manuscript, and Chris Baker, Nancy Kanwisher, Laura Lewis, and Michael Cohen for helpful discussion. We also thank Tommy Botch, Yeo Bi Choi, and Anna Mynick for assistance with data collection. This work was supported by a grant from NVIDIA to CER. A.S. is supported by the Neukom Institute for Computational Science.

## Author contributions

A.S. and C.E.R. conceived the idea and designed the research. A.S. collected the data. A.S. and M.M.B. analyzed the data. A.S. wrote the initial manuscript draft. A.S., E.H.S., and C.E.R. wrote the paper.

## Competing interests

The authors declare no competing interests.
