## [Peer Review File · Nature Communications]

Reviewers' Comments:

Reviewer #1:

Remarks to the Author:

Scene perception and Scene Memory

In this paper the authors conduct 4 studies and conclude that brain areas involved in scene memory are more anterior than brain areas that are involved in scene perception, arguing against prior results that have suggested that the same areas are involved in memory and perception.

I was excited to read this paper as I found the results intriguing. I also was impressed that the authors conducted multiple complimentary experiments on the same participants. While the experiments are interesting, after reading the paper in detail, I do not find that the data support the conclusions. The main reason for this assessment is that I believe that the authors have misidentified the regions involved in scene perception because of an in-optimal experimental design. Had they used a better localizer for place areas, I believe that they would have replicated results of prior studies that showed that similar regions are involved in place perception and place memory. Put another way, I believe that what is different between the present study and the past studies is how the place areas are defined, and not that different areas are involved in place memory and place perception.

Beyond the ROI definition - the data, in principle, could support a theory of a change of representation from being more stimulus based (vision) in the posterior parts of the CoS and PHG to more memory based in the more anterior aspects of the CoS and PHG (perhaps overlapping entorhinal and perirhinal cortices), but additional more fine grained analyses will be needed to support this interpretation.

Major concerns

1) In the present study, the place areas are defined by a contrast comparing brain responses to places vs. faces. This contrast is too liberal and identifies a cortical expanse spanning multiple brain regions rather than a 'place area' in each of their 3 anatomical foci of interest.

To illustrate this point, I will focus on the human ventral temporal cortex (VTC). The contrast of places vs faces, which the authors use to identify the PPA, is not selective enough to identify the PPA. To identify the PPA, the authors need to contrast activations to places vs multiple other stimuli, not just faces vs places.

In fact, the contrast faces vs places as used by the authors in the present study, identifies the medial half of human ventral temporal cortex (VTC), see white contour in Figures 1-3, of which the PPA is only a subregion (Weiner 2018, Weiner & Grill-Spector 2014). Looking at Figures 1-3, the region that the authors identify as the PPA spans about 5 cm in length and about 1-2 cm in width encompassing the entire medial aspect of VTC. This region cannot be a single brain area as it is larger than V1, which is known to be the largest visual area in the human brain (which surface area is about 3 cm, Dougherty 2003; Schwarzkopf 2011).

Further, many studies have shown that the region in medial VTC that the authors identify as the 'PPA', is not a single area but instead contains multiple retinotopic areas VO1/VO2/PHC1/PHC2 (Brewer 2005; Arcaro 2009). Not only is the medial VTC, not a single brain area, its selectivity to visual stimuli is not best characterized by its involvement in place perception. Many studies have shown that the medial VTC can be equally characterized by its preference to (i) peripheral vs foveal stimuli (Levy 2001; Weiner 2014), (ii) inanimate vs animate stimuli (Martin 1995) and (iii) large vs small objects (Konkle 2012). So, an equivalent title to the paper could be "regions involved in peripheral vision are more posterior than regions involved in place memory" or "regions processing large objects are more posterior than regions involved in place memory", which is different than the authors' take home message. Thus, to support their claims, the authors need to perform analyses in regions that show

higher specificity to places as well as analyse data in a much finer spatial grain.

Critically, examination of the hottest spots for memory or imagery in Figures 1-3 (as well as as in the supplements) suggests that what is labeled in the paper as the 'place memory area' seems to be similar to the most probable location of the PPA (Weiner 2018), which is different than the white contour labeled as the PPA in the present paper. So what seems to be reported as an anterior shift of responses during memory vs. perception could be due to a misidentification of the PPA; Put another way, I believe that had they had identified the PPA accurately, for example, by contrasting responses to places vs. many other stimuli that do not contain places, they would have found that there is substantial overlap between regions engaged in place perception and place imagery (as in O'Craven et al 2000) or place perception and place memory (as in Polyn et al 2005).

To alleviate my concern, the authors should compare their place perception and place memory regions to the most probably location of the PPA (Weiner 2018). This comparison needs to be done using single subject analyses and cortex based alignment. If they find that the hot spot for memory or imagery falls outside the most probable location of the PPA, it will verify their findings using an independent and accepted definition of the PPA. However, if they find that the memory/imagery region falls within this region, then it would argue against their findings.

I note that the same concerns, I raise for the PPA is valid for the place areas in the RSC (referred here as the MPA) and the TOS (referred here as the OPA). To identify the other place areas, they could run a localizer with many different categories and identify within each subject the actual place areas based on contrasting responses to places vs many other categories.

2) A related concern is that since the medial VTC is not a homogeneous region, and is composed of several regions, I do not believe it is justified to do analyses or stats on the mean response of this ROI as different voxels in this territory will likely have different response characteristics. The authors should either use accepted parcellations by retinotopy (Brewer 2005, Arcaro 2008; Wang 2014) or function/ functional connectivity (Baldassano 2012/2016; Glasser 2016) or analyze small anatomical sections from posterior to anterior in a systematic way to show how response characteristics change according to their hypothesized posterior to anterior axis (same comment for OPA/MPA).

3) The authors show the spatial relation of the regions using a center of mass analysis using regions defined under the different tasks using the same statistical threshold ($p < 0.001$), but many studies have shown that memory and imagery activations are generally weaker than activations to visual stimuli. It is possible that the most visually selective voxels to places are also the most involved in place memory or place imagery. Thus, I wonder if the pattern of results would change if the authors compared the center of mass of constant-sized regions defined for the different tasks. For example, the top 10% of visually selective voxels to places vs faces vs. the top 10% of voxels selective for place recall vs. face recall.

4) Limitations of functional connectivity analyses.

In Figure 2c, the authors report significantly higher correlations within network than between network during a movie viewing task. The data show that all the correlations are positive and low (range of 0.1-0.2). I wonder if the results depend on the choice of ROIs. Reiterating point (1) above, I do not think it is valid to examine the correlations among averaged time courses of large ROIs spanning several cms. Further, the regions outlined in perception are larger than the regions outlined in memory and the latter are a subset of the former. So, these regions are not independent from each other, and perhaps it is not surprising that the correlations are overall positive.

If the authors seek to use functional correlations to validate their functional results, they need to do

independent analyses on more spatially constrained regions with homogeneous selectivity. Further, they do not need to do these correlation analyses on a ROI level, but do them at the voxel level. For example, they could calculate the pairwise correlations among all voxels within the union of their red and white regions of their hypothesized memory and perception networks. From these pairwise correlations they could generate a cross-correlation matrix of all pairwise correlations among voxels; Applying clustering algorithms or using MDS on a distance version of this matrix they could examine what is the structure of the correlations. This would allow the authors to test in a data driven manner if voxels within network ('perception'/'memory') are more correlated than between networks.

A related question is why is the functional connectivity analysis is done only for the experiment comparing responses across familiar and unfamiliar places and not for the other experiments in which perception is compared to imagery or recall? It would be important to replicate this analysis in the unfamiliar video task and imagery task of experiment 4 to verify if this pattern of functional correlations results is task specific or task general.

5) Lack of justification for various experiments.

The writing of the results is very terse, and the choice of the various experiments and analyses is not well explained.

In experiment 1 the authors compare seeing scenes and faces vs recalling familiar faces and places

In experiment 2 the authors compared brain responses when subjects saw videos of familiar places vs unfamiliar places

In experiment 3 the authors examined the correlation between regions when subjects saw videos of unfamiliar places vs imaging familiar places

In experiment 4 the authors compared brain responses during viewing of unfamiliar scenes to imagery of familiar scenes.

Why were these particular tasks chosen? These tasks vary by multiple components, not only top-down attention as indicated by the authors, but also by task, familiarity, and the involvement of imagery. The author should expand the introduction to better explain the theoretical framework and how these experiments test various hypotheses.

6) The authors claim that they identify new place memory areas in the brain. It is unclear to me what aspect in their data supports the notion that the activation patterns reported here provide evidence for new brain areas.

Minor points

1) Citations: MPA was identified as RSC (e.g. Maguire 1998; Ino 2002; Epstein 2007); The OPA was identified by Levy 2001 and Hasson 2002;2003 as the TOS; Please cite the original findings and not the ones from a decade later

2) Figure 1, what are the units in the polar plot? mm?

3) The authors use place and scene interchangeably, but I wonder if they are the same. For example, a crowd of people could be a scene, but it is not a place.

Reviewer #2:

Remarks to the Author:

The study by Steel et al. shows a distinction between posterior and anterior scene selective areas, where the posterior parts of these areas (PPA, OPA and MPA/RSC) are involved in scene perception, and the more anterior parts are involved in scene memory. They show this both by contrasting fMRI activity for scene-vs-face perception and memory (experiment 1), and through direct comparison of scene perception and memory (experiment 4). They further show that the anterior scene memory regions are involved in both recall and recognition memory (experiment 2), and that the anterior scene memory regions are functionally connected to the hippocampus while the posterior scene

perception regions are functionally connected to the visual system, and both systems have stronger connectivity within them than between (experiment 3), supporting their proposed role in memory and perception respectively.

The paper is clearly and concisely written, and the experiments and analyses are well planned and executed – it seems that the authors took great care to rule out potential confounds (such as dropout artifacts and vividness), used both controlled and naturalistic stimuli, performed partial replication of their results, and showed that the effects are strong enough to appear in all individual subjects. The discussion is elegant and sheds light on the relevance of these findings to previous neuropsychological and cognitive results. Overall, I believe that the results of this paper are important to the field, as they combine several previous lines of research from the literature into a coherent framework, shed light on the possible basis of neurological impairments in scene perception and memory, and may have implications for understanding the relation between perception and memory beyond just the spatial domain.

My main comments below are about interpretation of the results in terms of distinction vs. overlap between the systems, and some missing discussion of relevant previous results from the literature.

Major comments:

1. The authors stress at several points in the manuscript that the scene-perception and scene-memory systems are distinct from one another, and have a discussion paragraph related to the implications of this distinction/dissociation. While the results convincingly show that activity “migrates” anteriorly in all scene regions when transitioning from perception to memory, I am less convinced of the full dissociation between different systems, as the results may also indicate a combined system with with gradual transition inside it. This is due to several reasons:

a. The authors show a significant overlap between the regions identified for perception and memory (especially in the PPA and MPA), although this overlap was only assessed qualitatively. It could be a good idea to assess this overlap quantitatively, and test whether there is a sharp transition between the systems (perception-overlapping-memory) or a gradual transition.

b. The connectivity analysis in experiment 3 used partial correlation with other correlated regions, which can remove the common variance / signals between the memory and scene regions. Therefore, the conclusion that the regions are not correlated in their activity across the networks seems too strong. In addition, while the differences in correlation of the two systems to the hippocampus and visual cortex appear striking, the difference of the between-network vs. within-network connectivity values appears to be driven only by the VMPA-MPA weak correlation and not by other regions (at least it seems so visually from the correlation matrix).

c. One important methodological issue is that the authors used in some of the analyses “constrained ROIs”. The methods describe “ROIs were constrained to unique members of the top 300 vertices from the perception and memory areas” – this is unclear – I assume this means that the 300 voxels with the strongest contrast values from experiment 1 were taken in each region? This is a crucial point, as 1) it is not described in the main text and this may be misleading, and 2) this might result in getting non-overlapping ROIs from the originally overlapping ones. Therefore, the systems might seem artificially dissociated when they are actually partially overlapping: this is very relevant for interpretation of results in experiment 4 (memory regions not involved in perception and vice versa) and in experiment 3 (connectivity between and within systems). If my reading of the methods is correct, please describe this clearly and add it also to the main text.

In my opinion, the issue of overlap does not affect the main conclusions of the paper or their importance, that would be equally valid if the systems are partially overlapping instead of distinct. But it is an issue that needs to be addressed and discussed, preferably with a formal measurement of the amount of overlap. If the systems are indeed partially overlapping, the analyses can also be performed on the parts of the two systems that are non-overlapping to show that parts of this system are dissociable while another part is common to both perception and memory.

2. The authors write in the abstract “we report a new network of brain areas... we reveal three cortical

areas... This new network..." and so on, and in the introduction they write "We henceforth refer to these previously undescribed regions as 'place-memory areas'". These claims of "finding new areas and networks" seems exaggerated. Subdivisions of the PPA to posterior and anterior parts involved in perception and memory respectively have been suggested by Epstein 2008 (TICS) and Baldassano et al. 2013 (NeuroImage), who also showed that these subregions have differential connectivity patterns. Baldassano et al. 2016 (eNeuro) also showed differential connectivity in anterior and posterior PPA and OPA (but not MPA) and suggested that these constitute distinct networks for vision and memory. Findings of differences in processing in MPA posterior-anterior subdivisions were described in Silson et al. 2019 (J Neuroscience), Burles et al. 2017 (Brain Struct Func) and 2018 (Front Hum Neurosci), and two papers by Chrastil et al. 2018 (both in Behavioral Neuroscience). In the present paper the authors convincingly combine these lines of research into a coherent framework that spans all scene regions, show the different results in the same subjects and at the single-subject level, and extend the knowledge of the scene-selectivity and type of memory operations these regions are involved in. However, the claims of finding a "new network of previously undescribed brain areas" seems overblown, as the anterior regions and their involvement in scene memory have been investigated and discussed before. I would suggest removing these claims and changing them to something more neutral (e.g. "we report a network... we identified three cortical areas..."). In addition, I think that a thorough discussion of the previous papers described above (or other relevant ones) and how they relate to the current findings is important to add.

Minor comments / suggestions:

1. In the abstract, the authors write "This new network, which conveys memory signals to visually responsive posterior cortex" – as far as I could tell, the authors didn't show that the anterior regions convey any signals to the visual regions, so this claim should be removed (although this could be a nice point for the discussion).
2. In experiment 1, the authors show that in face memory and perception there is no similar shift between adjacent posterior-anterior regions, and have an interesting discussion paragraph dedicated to not finding a "face memory network". However, looking at the individual subjects in the supplementary gif file, it seems that there is a region that is more active for face memory than perception – in the medial parietal lobe (in line with some of the authors' recent eLife paper). If this stands statistical analysis, it might be an interesting finding to briefly report or show in a supplementary figure (although it's not the main subject of the paper).
3. In experiment 2, the authors show that the scene memory regions responded more strongly to familiar vs. unfamiliar stimuli, and this was significantly stronger than in scene perception regions. But did scene perception regions by themselves also respond more strongly to familiar vs. unfamiliar stimuli? It seems so in Figure 2B. While this does not affect the conclusions, it is interesting in itself and can be easily tested.
4. The authors refer in experiment 1 to their memory task as "scene memory", and seem to refer to the exact same task instructions as "scene imagery" in experiment 4. I found this confusing – if there is a difference, please state it clearly; if there isn't, please use the same terminology to describe these results and tasks.
5. The authors discuss the scene memory regions as being part of the default-mode network, but they don't actually show this anywhere besides them being in the same general parts of the brain. This overlap can be easily measured by using templates from the literature or by extracting the DMN from their own movie-watching data, although this can also be solved by saying something like "these seem to lie in the same parts of the cortex... future studies directly comparing these systems could help elucidate this issue".
6. General organization suggestions – I believe that readers will benefit from a summary of the main results of all of the experiments somewhere in the paper, as it is currently difficult to create a mental

overview of the paper's results without careful reading and note-taking through all of it. In addition, some of the figures – particularly figure 2 – might benefit from subheadings in the figure, or an in-figure legend of abbreviations (e.g. adding region abbreviations next to the appropriate cortical regions in Figure 2A), as the number of abbreviations used is slightly overwhelming and requires going back-and-forth to the legend to understand them.

7. Supplementary figure 1 has a very low resolution – it would be better if the resolution could be increased in the final version.

Reviewer #3:

Remarks to the Author:

Steel and colleagues report a series of 4 fMRI experiments in which they identified and characterized three cortical areas that contribute preferentially to place memory in comparison to scene perception. As demonstrated in a group of 14/13 participants, these areas are localized on the lateral, ventral and medial cortical surfaces, sit immediately anterior to brain regions that are preferentially involved in scene perception (i.e. the PPA, OPA, and MPA), and demonstrate greater functional connectivity with each other in comparison to scene perception regions, pointing towards a cortical network for place memory.

The interface between memory and perception has been the focus of increasing interest over the past 10 - 15 years, and this study is, therefore, both timely and interesting. The inclusion of multiple experiments as well as the general convergence of findings across these experiments is a strength, and overall, the paper is clearly written. My major concern, however, is that the design of the experiments and the consideration of the findings lack the necessary finesse/precision to shed light on the functional and theoretical importance of these 'place memory' areas. In other words, while this study reports an interesting finding in the localization of these areas, the interpretation of the reported activity and its conceptual significance, at least as the findings are currently reported, is not clear.

1) The current study somewhat glosses over the heterogeneity and complexity of 'place memory'. For instance, memory age (childhood vs. adulthood) and type (e.g. semantic vs. episodic) have been suggested to impact the neural correlates underlying place memory (e.g. Moscovitch/Nadel and colleagues, Squire and colleagues) and yet these factors were not taken into account when the place stimuli lists were created for each participant (e.g. participants could recall a childhood kitchen and/or a current kitchen), or when the data were analyzed. Is there any additional insight that the authors could provide into this issue, for instance, by providing additional details on the average age of the familiar memories and whether there was significant variance across participants?

2) Related to the above, further details on the types of familiar and unfamiliar scene stimuli that were used would be helpful for the interpretation of the data. For instance, were there similar categories/types and numbers of different categories/types of scenes for the familiar and unfamiliar stimuli (e.g. indoor vs. outdoor, cluttered vs. uncluttered scenes, etc) across the different experiments?

3) One interesting finding is the response of the hippocampus to the viewing of familiar vs. unfamiliar panning movies, and moreover, the strong correlation between place-memory area and hippocampal activity. These data point towards a role for the hippocampus in place memory and appear somewhat discrepant to studies that have implicated the hippocampus in both perception and memory (e.g. work of Maguire group in the context of scene construction, Lee/Barens in the context of spatial feature conjunction processing). Indeed, there are some who would perhaps suggest that the hippocampus could function as an interface between scene perception and memory, and that this structure could subservise representations/processes that support both perception and memory. Discussion of this issue is important - for example, do the authors consider the observed lack of hippocampal involvement

during perception a function of their task design or support for the notion that the hippocampus contributes to memory only? Similarly, some have argued for a perceptual and mnemonic role for the perirhinal cortex for objects/faces, and this region did not feature in the face-related findings for Experiment 1.

4) I would appreciate a little more detail and discussion regarding the actual location (in terms of mean group max coordinates, anatomical structures) of the different memory place areas. For example, does the VPMA extend into parahippocampal cortex and more anteriorly into entorhinal cortex, areas that have been implicated in spatial/contextual memory and navigation? Similarly, how does the MPMA relate to the retrosplenial regions that have been implicated in navigation, and episodic/autobiographical memory? Indeed, thinking more broadly, how do the authors view their place memory area network in relation to the network of regions that is typically associated with spatial navigation and spatial memory?

5) The authors state that their findings are inconsistent with previous fMRI work that has suggested overlapping neural substrates for perception and mental imagery of high-level stimuli. According to my knowledge, however, in many previous fMRI studies on perception vs. imagery, participants were asked to perceive and visualize the same stimuli. This contrasts to the current study in which participants perceived unfamiliar places but imagined personally familiar places. Thus, not only was there a difference in memory (unfamiliar vs. familiar) but also stimulus content, and it is unclear which of these is underlying the observed discrepancy with previous work. I think the evidence (in particular Expt 2, Extended data Fig 7) suggests that memory is the key factor here but perhaps the most convincing way to demonstrate this empirically would be to scan participants while they perceive unfamiliar scenes, allow them to then become highly familiar with the same scenes, and then scan them again when they are asked to imagine or see the same scenes again - an anterior shift in activity would provide further weight to the authors conclusions.

6) Page 5 states that in Experiment 3 "participants watched a movie designed to concurrently engage perceptual and mnemonic representations of complex real-world environments". I may have misunderstood something but this statement confused me since the Methods states that for the naturalistic movie, "Participants were not asked whether places were familiar to them". Given this latter statement, how can the authors be certain that the naturalistic movie engaged both perceptual and mnemonic representations?

7) Apologies if I missed this but I assume that the ROIs used for Experiment 3 were mutually exclusive in their extent (i.e. the neighbouring memory and perception ROIs that were used to calculate correlations in activity were not overlapping).

8) Were the same stimuli repeated across experiments 1, 2, and 4? For instance, were the same unfamiliar panning movies used in Experiment 2 as in Experiment 4 (in which case, they may not have been as unfamiliar in Experiment 4 given the same participants would have seen them before?).

9) A very minor point (and perhaps I just missed this) but it would be useful to describe the collection of the data for Extended data Figure 2 in the Methods section of the main manuscript.

REVIEWER COMMENTS

Reviewer #1 (Remarks to the Author):

Scene perception and Scene Memory

In this paper the authors conduct 4 studies and conclude that brain areas involved in scene memory are more anterior than brain areas that are involved in scene perception, arguing against prior results that have suggested that the same areas are involved in memory and perception.

I was excited to read this paper as I found the results intriguing. I also was impressed that the authors conducted multiple complimentary experiments on the same participants. While the experiments are interesting, after reading the paper in detail, I do not find that the data support the conclusions. The main reason for this assessment is that I believe that the authors have misidentified the regions involved in scene perception because of an in-optimal experimental design. Had they used a better localizer for place areas, I believe that they would have replicated results of prior studies that showed that similar regions are involved in place perception and place memory. Put another way, I believe that what is different between the present study and the past studies is how the place areas are defined, and not that different areas are involved in place memory and place perception.

Beyond the ROI definition - the data, in principle, could support a theory of a change of representation from being more stimulus based (vision) in the posterior parts of the CoS and PHG to more memory based in the more anterior aspects of the CoS and PHG (perhaps overlapping entorhinal and perirhinal cortices), but additional more fine grained analyses will be needed to support this interpretation.

We appreciate the reviewer's enthusiasm for our study and their excellent suggestions for additional analyses to strengthen our conclusions. In particular, we thank the reviewer for raising the important point regarding the definition of the scene-perception areas. In light of the reviewer's comment, we have reanalyzed our data using the reviewer's suggested methods for defining the scene perception areas, and we find that our results are unchanged. These analyses are described in detail below. These suggestions, along with other thoughtful control analyses suggested by the reviewer, have greatly strengthened our manuscript. We hope that the reviewer finds their concerns appropriately addressed through these revisions.

Major concerns

*1) In the present study, the place areas are defined by a contrast comparing brain responses to places vs. faces. **This contrast is too liberal** and identifies a cortical expanse spanning multiple brain regions rather than a 'place area' in each of their 3 anatomical foci of interest.*

*To illustrate this point, I will focus on the human ventral temporal cortex (VTC). The contrast of places vs faces, which the authors use to identify the PPA, is not selective enough to identify the PPA. **To identify***

the PPA, the authors need to contrast activations to places vs multiple other stimuli, not just faces vs places.

The reviewer raises several important points in Major Comment 1, which we will address point-by-point in line below, breaking Comment 1 into Comments 1a-e.

1a) The reviewer suggests that the contrast used to define scene-perception areas in our study (scenes > faces) is too liberal. They suggest that this liberal contrast might inflate the size of scene areas, and possibly contribute to a more posterior neuroanatomical location of the ventral scene area (PPA). We thank the reviewer for raising this important point, and for suggesting ways to address their concern: “contrast activations to places vs multiple other stimuli, not just faces vs places.” Based on the reviewer’s suggestion, we reanalyzed our data using a more conservative contrast to define PPA. Specifically, because we included manipulable objects (e.g. paper clip, hammer, tennis racket, etc.) as a condition in our perceptual category-localizer, we could use objects as an additional category for comparison when localizing PPA (scenes > faces + objects). As the reviewer predicted, this new contrast did reduce the size of our scene perception areas. Yet, importantly, the perception areas were still consistently located posterior to memory areas in all subjects. We describe these analyses in details below.

First, we tested whether the location of the scene-perception areas changed significantly when we used the more conservative contrast. We compared the weighted centers-of-mass for each scene perception area as defined using both the more liberal (scenes > faces) and more conservative (scenes > faces + objects) contrasts. We did not observe a significant shift in the weighted center-of-mass of MPA (Left hemisphere: $t(13) = 0.37$, $p = 0.71$, Right hemisphere: $t(13) = 1.65$, $p = 0.12$) or PPA (Left hemisphere: $t(13) = 0.99$, $p = 0.34$, Right hemisphere: $t(13) = 0.96$, $p = 0.35$). Interestingly, the weighted center-of-mass of OPA was shifted significantly *posteriorly* when objects were included in the contrast (Left hemisphere: $t(13) = -2.87$, $p = 0.013$, Right hemisphere: $t(13) = -2.22$, $p = 0.045$), and thus the more conservative contrast suggested by the reviewer would have made it *more likely* to observe an anterior shift of place-memory relative to scene-perception activity.

Second, we re-analyzed the location of place-memory areas as compared with scene-perception areas that were defined using the more conservative contrast (scenes > faces + objects) (Response Figure 1). This analysis replicated our initial result: on all surfaces, the place memory area was shifted anteriorly relative to the scene-perception area when objects were included in the localizer contrast (Medial surface -- Left hemisphere: $t(13) = 7.00$, $p < 0.0001$, Right hemisphere: $t(13) = 5.24$, $p = 0.00016$; Ventral surface -- Left hemisphere: $t(13) = 10.27$, $p < 0.0001$, Right hemisphere: $t(13) = 6.97$, $p < 0.0001$; Lateral surface -- Left hemisphere: $t(13) = 16.67$, $p < 0.0001$, Right hemisphere: $t(13) = 9.89$, $p < 0.0001$).

All in all, as the reviewer suggests, a more conservative contrast does reduce the size of our scene perception areas. However, we observe a significant anterior-shift for place-memory as compared with scene-perception activity in all individual subjects, regardless of whether scene perception areas are defined using a one or multiple categories as the contrast. We think that this new analysis strengthens

the conclusion of our manuscript and have included it in the Extended data (including Extended Data figure 2) and added this analysis to the Main Text (Page 3, Lines 90-93). However, we continue to use our original contrast (scenes versus faces) in the Main Text to be most analogous to the contrast that we used during memory recall (places versus people).

Response figure 1. Contrasting scenes versus faces and objects (instead of scenes versus faces as in Figure 1) does not impact the location of scene-perceptual areas. Left. Four example participants demonstrate the minimal impact of including visually objects when defining PPA. PPA defined by comparing blocks of scenes versus faces (used for Experiment 1) is outlined in black, and the PPA defined by comparing blocks scenes versus faces and objects is shown in hot colors. Right. Polar plots comparing scene-perception and place-memory areas. The center of the polar plot refers to the weighted center-of-mass of the scene-selective regions defined by contrasting scenes versus faces and objects in each participant, and each data point refers to the weighted center-of-mass of the place-memory area. For all subjects, the center-of-mass of the place-memory area was significantly anterior to the weighted center-of-mass of the scene-perception area on all surfaces (all ps < 0.001). The polar plots are ordered with ventral (top), lateral (middle), and medial (bottom). Anterior is towards the top of the figure for all surfaces/polar plots. Polar plot units are millimeters.

1b) In fact, the contrast faces vs places as used by the authors in the present study, identifies the medial half of human ventral temporal cortex (VTC), see white contour in Figures 1-3, of which the PPA is only a subregion (Weiner 2018, Weiner & Grill-Spector 2014). Looking at Figures 1-3, the region that the authors identify as the PPA spans about 5 cm in length and about 1-2 cm in width encompassing the entire medial aspect of VTC. This region cannot be a single brain area as it is larger than V1, which is

known to be the largest visual area in the human brain (which surface area is about 3 cm, Dougherty 2003; Schwarzkopf 2011).

1b) The reviewer expressed concern regarding the large size of PPA depicted in Figure 1. We appreciate the reviewer raising this point, which allows us the opportunity to clarify our methodology that took this issue into account.

Figure 1, indeed, shows large ROIs composed of data from all nodes that survived correction at a vertex-wise threshold of $p < 0.001$. However, these liberally-thresholded maps are for illustrative purposes only and are not used in subsequent analyses. Crucially, for Experiments 2-4, we considered only unique members of the top 300 most scene-perception and place-memory selective surface vertices. This left us with functionally homogenous regions for comparison, rather than areas that covered larger swaths of cortex. We hope that this important point of clarification alleviates the reviewer's concerns (here and below) have updated the manuscript to make the ROI definition (including the constrained vertices) clearer to readers in light of the reviewer's comment.

To ensure that we evaluated functionally homogenous regions and to control for differences in ROI sizes across regions, we constrained the scene-perception and place-memory areas to the unique members of the top 300 most scene-perception/place-memory preferring vertices for all subsequent experiments. (Page 6, Lines 178-182)

Please see also our response to Major Comment 3 below, which shows that the anterior-bias of memory as compared with perception regions observed in Experiment 1 is preserved when only the top 300 most scene-perception and place-memory selective surface vertices are considered.

1c) Further, many studies have shown that the region in medial VTC that the authors identify as the 'PPA', is not a single area but instead contains multiple retinotopic areas VO1/VO2/PHC1/PHC2 (Brewer 2005; Arcaro 2009). **Not only is the medial VTC, not a single brain area, its selectivity to visual stimuli is not best characterized by its involvement in place perception.** Many studies have shown that the medial VTC can be equally characterized by its preference to (i) peripheral vs foveal stimuli (Levy 2001; Weiner 2014), (ii) inanimate vs animate stimuli (Martin 1995) and (iii) large vs small objects (Konkle 2012). So, an equivalent title to the paper could be "regions involved in peripheral vision are more posterior than regions involved in place memory" or "regions processing large objects are more posterior than regions involved in place memory", which is different than the authors' take home message. Thus, to support their claims, the authors need to **perform analyses in regions that show higher specificity to places as well as analyse data in a much finer spatial grain.**

1c) The reviewer raises the theoretical concern that activity in medial VTC may not be best characterized as "place-selective," and, in particular, notes that non-categorical gradients have been proposed to describe VTC organization, including eccentricity, real-world size, and animacy. While we agree with the reviewer that these gradients may describe the broad organization of VTC, we think that prior neuropsychological and neuroimaging literature strongly support the existence of a local scene-selective area in medial VTC within this broader organization. This is for three reasons. First, patients with lesions

in medial VTC can exhibit landmark agnosia and place-specific recognition deficits (Aguirre 1999), while leaving recognition of other classes of objects intact. Second, neuroimaging experiments mapping object-selective activity within PPA have found its activation to large objects weak and sparse compared to its preference to scenes (Julian, Ryan, and Epstein 2017). Finally, the large objects/non-manipulable objects used to characterize real-world size preference are often either 1) strongly associated with places (e.g. sofa with living room) or 2) can be considered a “place”, in the sense that a person can enter them (e.g. a pagoda) (Konkle and Oliva 2012; He et al. 2013). Together, this is strong evidence that a scene-selective area exists in medial VTC, although we agree with the reviewer that this area lies within broader organizational gradients of VTC (Grill-Spector and Weiner, 2014). Therefore, in the revised manuscript, we continue to use the term “scene-perceptual areas” because we define them using a category localizer (scenes versus faces (+objects)), rather than another perceptual organizing principle (e.g., foveal versus peripheral).

We also note that the proposed gradients the reviewer raises are only relevant to the medial-lateral axis of VTC, while our paper is principally concerned with the anterior-posterior axis. As such, we do not believe that the broad organizing principles of the medial-lateral axis (real-world size, eccentricity, or animacy) impinges on our main conclusions: that memory-related place activity falls anterior to visually related scene activity in medial VTC. However, we agree with the reviewer that discussing organizational principles of the ventral temporal cortex would add important context to our results. Therefore, we have added the following to the discussion section of our manuscript.

The underlying organizational principles of ventral temporal cortex have been hotly debated. Along the medial-lateral axis, multiple organizational gradients have been identified, including eccentricity^{24,26,40}, animacy⁵¹⁻⁵⁴, and real-world size⁵⁵. On the other hand, the anterior-posterior axis has been conceptualized as a gradient from concrete/perceptual representations to abstract/conceptual representations^{40,46,56-61}. [...] (Page 14, Lines 366-381)

Finally, we have performed two new analyses relevant to the concern regarding the specificity of our ROI to places and the spatial scale of our analysis: 1) we analyze our data considering a higher specificity to scenes compared to other categories (Response 1a) and 2) we compare the location of the top 300 most scene-perception and place-memory selective vertices (Response 3 below). As detailed in these responses, our conclusions are robust even when drawn from the most scene-selective ROIs at a fine spatial scale.

1d) Critically, examination of the hottest spots for memory or imagery in Figures 1-3 (as well as in the supplements) suggests that what is labeled in the paper as the ‘place memory area’ seems to be similar to the most probable location of the PPA (Weiner 2018), which is different than the white contour labeled as the PPA in the present paper. So what seems to be reported as an anterior shift of responses during memory vs. perception could be due to a misidentification of the PPA; Put another way, I believe that had they had identified the PPA accurately, for example, by contrasting responses to places vs. many other stimuli that do not contain places, they would have found that there is substantial overlap

between regions engaged in place perception and place imagery (as in O'Craven et al 2000) or place perception and place memory (as in Polyn et al 2005).

To alleviate my concern, the authors should compare their place perception and place memory regions to the most probable location of the PPA (Weiner 2018). This comparison needs to be done using single subject analyses and cortex based alignment. If they find that the hot spot for memory or imagery falls outside the most probable location of the PPA, it will verify their findings using an independent and accepted definition of the PPA. However, if they find that the memory/imagery region falls within this region, then it would argue against their findings.

1d) We thank the reviewer for this suggestion to use a widely accepted probabilistic definition of PPA as a stable reference point against which to compare place memory activity. Based on the reviewer's comment, we compared the weighted center-of-mass of the ventral place memory area to the center of the most probable location of PPA from Weiner et al. 2018. We found that the ventral place memory area was significantly anterior to this definition of PPA, verifying our previous result (Left hemisphere: $t(13) = 11.82$, $p < 0.001$, Right hemisphere: $t(13) = 9.37$, $p < 0.001$; Response Figure 2).

This analysis makes our results much clearer by showing that the ventral place memory area falls anterior to a widely accepted, conservative definition of PPA. We have therefore included this result in the Main Text (Page 6, Lines 90-93) and as a figure in our Extended data (Extended data figure 3).

Response Figure 2. The VPMA is located anterior to the probabilistically-defined PPA. Left. Data from an example participant. The place memory area (hot colors) is anterior to the most probable location of

PPA (White; Weiner et al. 2018). The dashed line shows the participant's original PPA defined using the contrast of scenes versus faces. Right. Polar plot showing the center-of-mass of the most probable location of PPA (Polar plot center) compared to the weighted center of mass of the ventral place memory area (data points). These findings are consistent with our results using individually localized PPA (Main Text Figure 1): the center of mass of the ventral place memory area is anterior to the probabilistically defined PPA in all participants.

1e) I note that the same concerns, I raise for the PPA is valid for the place areas in the RSC (referred here as the MPA) and the TOS (referred here as the OPA). To identify the other place areas, they could run a localizer with many different categories and identify within each subject the actual place areas based on contrasting responses to places vs many other categories.

We thank the reviewer for raising this important concern regarding the definition of the scene-perception areas. Because of its central importance, we have responded to this comment above (Please see Response 1a).

*2) A related concern is that **since the medial VTC is not a homogeneous region, and is composed of several regions, I do not believe it is justified to do analyses or stats on the mean response of this ROI** as different voxels in this territory will likely have different response characteristics. The authors should either use accepted parcellations by retinotopy (Brewer 2005, Arcaro 2008; Wang 2014) or function/functional connectivity (Baldassano 2012/2016; Glasser 2016) or analyze small anatomical sections from posterior to anterior in a systematic way to show how response characteristics change according to their hypothesized posterior to anterior axis (same comment for OPA/MPA).*

We thank the reviewer for these suggestions. As also described in our Response 1b above, we shared the concern that the large swaths of cortex shown in Figure 1 may not be functionally homogenous. These liberally-defined regions are shown for illustrative purposes in Figure 1, and our analyses/stats are not based on the mean response of these ROIs. Rather, we constrained the analyses in Experiments 2-4 to the unique members of the 300 most scene-perception/place-memory selective vertices on the surface. This procedure isolated substantially smaller, functionally homogenous regions: the most perceptual/memory-recall selective regions. We have clarified this important methodological point in the Main Text in our MS. (Page 6, Lines 177-182)

In addition, the reviewer raises an interesting point with respect to the relationship between the place-memory regions and cortical retinotopic areas. Examining the overlap would provide more concrete evidence that the place-memory areas are functionally dissociable from scene-perception regions, and visual areas generally. To examine this question, we calculated the overlap between the ventral and lateral place memory areas and the most-probable locations of the retinotopic maps defined by Wang et al. 2014 (max probability at each vertex; Response figure 3). This data showed a striking differentiation between the retinotopic cortical areas and the place-memory areas. The overwhelming majority of the

place-memory selective vertices fell outside of retinotopic cortex on both the ventral and lateral surfaces. On the ventral surface, the place-memory selective nodes were largely anterior of retinotopic cortex, while the place-memory area on the lateral surface was in the caudal inferior parietal lobule encircled by, but not overlapping with, the retinotopic areas.

Response figure 3. Minimal overlap between cortical retinotopic maps and the place memory areas. The maximum probability projection of cortical retinotopic maps from Wang et al. 2014 are overlaid on one individual subject's data. The place-memory areas on the ventral surface fall largely anterior to the retinotopic regions PHC1 and PHC2. On the lateral surface, the place-memory area primarily fell outside of retinotopic cortex. Inset. Group-average pie charts showing the proportion of surface vertices of the place memory areas contained within each retinotopic region. Across all participants, the majority of vertices fell outside of the cortical retinotopic maps.

In addition, in response to Reviewer 3, we performed a group analysis of the place-memory localizer data and compared the peak of place-memory activity with the probabilistic retinotopic maps and found virtually no overlap between the place-memory activity and retinotopic maps.

Response Figure 4. Group-defined place-memory network activity falls outside of retinotopic cortex. Hot colors show the areas showing significant place-memory preferring responses, and outlined regions show maximum probability projection of the probabilistic retinotopic maps (Wang et al. 2014).

We think that this analysis provides greater detail regarding the location of the place-memory areas and have therefore discussed this finding in the Main Text and included these data as both a main figure (Figure 2a) and in the Extended data (Extended data figure 5).

We performed two additional analyses to characterize the cortical locations of the place memory areas. First, we asked: do the locations of the place-memory areas fall in retinotopic, or non-retinotopic cortex? We hypothesized that, unlike scene perception areas, which process visual input, place memory areas would lie in non-retinotopic cortex. To test this, we performed a group analysis of the place memory localizer and compared the peak activity for each area to retinotopic parcels based on a probabilistic atlas⁴². Strikingly, on all cortical surfaces, place-memory areas were consistently located anterior to the cortical retinotopic maps (Figure 2a) with minimal overlap (Extended data Fig. 5). This location in non-retinotopic cortex is potentially consistent with their role as preferentially mnemonic, rather than visual, areas. (Page 5, Lines 124-132)

3) The authors show the spatial relation of the regions using a center of mass analysis using regions defined under the different tasks using the same statistical threshold ($p < 0.001$), but many studies have shown that memory and imagery activations are generally weaker than activations to visual stimuli. It is possible that the most visually selective voxels to places are also the most involved in place memory or place imagery. Thus, I wonder if the pattern of results would change if the authors compared the center of mass of constant-sized regions defined for the different tasks. For example, the top 10% of visually selective voxels to places vs faces vs. the top 10% of voxels selective for place recall vs. face recall.

We thank the reviewer for this suggestion. Importantly as detailed above (Responses 1b and 2), we controlled for ROI size by thresholding to the top 300 most scene-perception/place-memory selective vertices in Experiments 2-4. But, crucially, we did not test whether the constrained place-memory activation was anterior to the scene-perception activation in Experiment 1.

Therefore, in light of the reviewer's comment, we repeated our topological analysis using the top 300 most scene-perception and place-memory selective surface vertices to confirm that the anterior bias was still present at the strictest threshold (Response figure 4). This analysis recapitulated our initial results using the full ROIs ($p < 0.001$ thresholded regions): there was a significant anterior shift from scene-perception to place-memory on all cortical surfaces in both the left and right hemisphere (Lateral - left hemisphere: $t(13) = 9.35$, $p < 0.0001$, right hemisphere: $t(13) = 8.62$, $p < 0.0001$; Ventral - left hemisphere: $t(13) = 4.84$, $p = 0.0003$, right hemisphere: $t(13) = 7.45$, $p < 0.0001$; Medial - left hemisphere: $t(13) = 2.86$, $p = 0.013$, right hemisphere: $t(13) = 2.36$, $p = 0.034$).

Response figure 4. Anterior shift of place-memory relative to scene-perception remains consistent after constraining to maximally selective surface vertices. Across the lateral, medial and ventral surfaces, we compared the weighted center-of-mass of both the scene-perception and place-memory selective areas after constraining the regions to the top 300 most category-selective surface vertices. This revealed a significant anterior shift in all cortical surfaces (all p s < 0.05). Units for polar plots are millimeters.

We sincerely appreciate the reviewer's suggestion, which addressed an important gap in our initial analysis. We now discuss this finding in the Main Text (Page 6-7, lines 177-182) and have added this figure to the Extended data (Extended data figure 7).

4a) Limitations of functional connectivity analyses.

*In Figure 2c, the authors report significantly higher correlations within network than between network during a movie viewing task. The data show that all the correlations are positive and low (range of 0.1-0.2). I wonder if the results depend on the choice of ROIs. Reiterating point (1) above, **I do not think it is valid to examine the correlations among averaged time courses of large ROIs spanning several cms.** Further, the regions outlined in perception are larger than the regions outlined in memory and the latter are a subset of the former. So, these regions are not independent from each other, and perhaps it is not surprising that the correlations are overall positive.*

*If the authors seek to use functional correlations to validate their functional results, **they need to do independent analyses on more spatially constrained regions with homogeneous selectivity.** Further, they do not need to do these correlation analyses on a ROI level, but do them at the voxel level. For example, they could calculate the pairwise correlations among all voxels within the union of their red and white regions of their hypothesized memory and perception networks. From these pairwise correlations they could generate a cross-correlation matrix of all pairwise correlations among voxels; Applying clustering algorithms or using MDS on a distance version of this matrix they could examine what is the structure of the correlations. This would allow the authors to test in a data driven manner if voxels within network ('perception'/'memory') are more correlated than between networks.*

4a) The reviewer notes that the correlation values in the network analyses presented in Exp 3 are low, and wonders if this depends on our choice of ROIs. We appreciate this question, as it offers the chance to clarify our methodology. Our connectivity measurement is the result of a partial correlation analysis, rather than a full correlation analysis; thus, correlation values are low, and should be interpreted as the correlation between two regions whilst controlling for the time course of BOLD activity in all other regions. We have opted for partial correlation because all regions are being driven by a strong bottom-up visual stimulus. In light of the reviewer's comment, we have added the following to the results section to clarify this point.

We analyzed the co-fluctuation of activity among the scene-perception and place-memory areas during the movie by calculating the correlation between the activity timeseries for each region pair, whilst partialling out the activation of all other regions (Fig. 4a). **Accounting for the activation of all other regions allowed us to assess the co-fluctuation of these areas above-and-beyond their shared response to the video stimulus, which causes all regions to exhibit overwhelming positively correlated activity.** (Page 9, Lines 244-249)

Regarding our choice of ROIs: we apologize for our lack of clarity - as noted above in Responses 1b and 2, for the analyses in Experiments 2-4, including the functional connectivity analysis (Experiment 3), we constrained our analysis to the unique members of the top 300 scene-perception and place-memory selective vertices to control for this issue. Constraining the regions leads to small, more functionally homogenous regions. With that in mind, we think the reviewer's suggested fine-grained voxel-level analysis of functional connectivity might stem from their misunderstanding of our original analysis.

4b) A related question is why is the functional connectivity analysis is done only for the experiment comparing responses across familiar and unfamiliar places and not for the other experiments in which perception is compared to imagery or recall? It would be important to replicate this analysis in the unfamiliar video task and imagery task of experiment 4 to verify if this pattern of functional correlations results is task specific or task general.

4b) We believe that the reviewer's concern here might stem from a misunderstanding of the stimuli presented in Experiment 3 (connectivity analysis). To clarify, the connectivity analysis in Experiment 3 is done on data collected while participants viewed an 11-minute naturalistic video (Manuscript Page 8, Lines 238-243; Page 20, Lines 581-595), not the familiar/unfamiliar panning videos of places used in Experiments 2 and 4. The naturalistic video in Experiment 3 included segments depicting a series of concatenated architectural tours, college admissions videos (including Dartmouth), and real estate advertisements. We chose this stimulus for two reasons: 1) it provided a naturalistic (task-neutral) state to test whether the scene-perception and place-memory areas formed functionally dissociable networks during visual processing, and 2) the long, 11-min scan provided ample data for a robust connectivity analysis. In the future, it would be interesting to look at whether the functional connectivity of the

place-memory and scene-perception areas changes as a function of task conditions or stimulus familiarity.

5) Lack of justification for various experiments.

The writing of the results is very terse, and the choice of the various experiments and analyses is not well explained.

In experiment 1 the authors compare seeing scenes and faces vs recalling familiar faces and places

In experiment 2 the authors compared brain responses when subjects saw videos of familiar places vs unfamiliar places

In experiment 3 the authors examined the correlation between regions when subjects saw videos of unfamiliar places vs imaging familiar places

In experiment 4 the authors compared brain responses during viewing of unfamiliar scenes to imagery of familiar scenes.

Why were these particular tasks chosen? These tasks vary by multiple components, not only top-down attention as indicated by the authors, but also by task, familiarity, and the involvement of imagery. The author should expand the introduction to better explain the theoretical framework and how these experiments test various hypotheses.

We thank the reviewer for impressing upon us the need to expand the discussion of our theoretical framework and the methods we employed in this study. To address their concern, we have added the following overview of our experiments at the end of Introduction:

Here, we sought to describe the neural basis of perceptually- and mnemonically-driven representations of real-world scenes in the human brain. To do this, we conducted a series of experiments using fine-grained, individual-subject fMRI to assess whether these responses are co-localized in the brain (Experiment 1). Surprisingly, this experiment revealed strong evidence to the contrary: in all subjects, visually recalling real-world locations evoked activity in three previously undescribed areas in posterior cerebral cortex, each lying immediately anterior to one of the three regions of the scene-perception network. We next characterized the functional and network properties of these “place-memory areas” (Experiments 2-3) and explicitly tested the wide-spread view that the neural substrates of perception and memory recall (i.e., mental imagery) activate shared regions of the human brain (Experiment 4). Together, our results reveal a network of brain areas that collectively bridge the scene perception and spatial memory systems of the human brain and may facilitate integration of the local visual environment with spatial memory representations. (Page 2-3, Lines 57-69)

We made similar additions throughout the Main Text and expanded various transition points between the experiments to better motivate their selection, including:

Page 7, Lines 184-189

Page 8, Lines 229-238

Page 10, Lines 289-297

Also, we note that the reviewer's description of Experiment 3 is not correct, as discussed above in 4a and 4b.

6) The authors claim that they identify new place memory areas in the brain. It is unclear to me what aspect in their data supports the notion that the activation patterns reported here provide evidence for new brain areas.

We argue that our data present strong evidence for the existence of the place-memory network in posterior cerebral cortex and thank the reviewer for the opportunity to clarify our basis for this claim.

In our paper, we apply four tests to establish the presence of the place memory areas, and to test whether they are distinct functional areas from the nearby scene-perception areas:

- 1) Are they anatomically dissociable from nearby scene-perception areas?
- 2) Is their functional profile distinct from nearby scene-perception areas (more readily characterized as driven by mnemonic versus perceptual tasks)?
- 3) Do they operate together as a distinct functional network during naturalistic conditions (more correlated with each other than with scene-perception areas)?
- 4) Do they show distinct functional connectivity profiles from nearby scene-perception areas (greater relation to memory structures, as compared with visual regions)?

All of these tests were passed. To restate our findings, the place memory areas are spatially dissociable from scene perception areas (Exp 1). They are also functionally dissociable from scene perception areas: the place memory areas show a distinct preferences for memory-related tasks including recognition memory (Exp 2) and visual memory recall (Exp 4). Finally, the place memory areas form a distinct functional network which, in contrast to the scene-perception areas, affiliates closely with memory structures in the brain during naturalistic viewing (Exp 3). These strong anatomical, functional, and network-level dissociations between the place-memory and scene-perception areas support the idea that these regions are separable. Importantly, place-memory areas can be readily identified in individual participants with large effect sizes and are internally replicated (Exp 1, Extended data Fig 1a), highlighting the generalizable and robust nature of our findings.

Minor points

1) Citations: MPA was identified as RSC (e.g. Maguire 1998; Ino 2002; Epstein 2007); The OPA was identified by Levy 2001 and Hasson 2002;2003 as the TOS; Please cite the original findings and not the ones from a decade later

We have added these citations to our manuscript (Page 2, Lines 48-49).

2) Figure 1, what are the units in the polar plot? Mm?

The units are millimeters. This information is presented in the figure legend and we have added this information to the figure caption as well.

3) The authors use place and scene interchangeably, but I wonder if they are the same. For example, a crowd of people could be a scene, but it is not a place.

Throughout the manuscript we use the word “scene” to refer to visually presented stimuli and we use the word “place” when referring to recalled stimuli. We now use the word “location” to describe a place in the real-world, which could be either visually presented or recalled. We have ensured that we use these terms consistently throughout the manuscript and thank the reviewer for flagging this important point of clarification.

Reviewer #2 (Remarks to the Author):

The study by Steel et al. shows a distinction between posterior and anterior scene selective areas, where the posterior parts of these areas (PPA, OPA and MPA/RSC) are involved in scene perception, and the more anterior parts are involved in scene memory. They show this both by contrasting fMRI activity for scene-vs-face perception and memory (experiment 1), and through direct comparison of scene perception and memory (experiment 4). They further show that the anterior scene memory regions are involved in both recall and recognition memory (experiment 2), and that the anterior scene memory regions are functionally connected to the hippocampus while the posterior scene perception regions are functionally connected to the visual system, and both systems have stronger connectivity within them than between (experiment 3), supporting their proposed role in memory and perception respectively.

*The paper is clearly and concisely written, and the experiments and analyses are well planned and executed – it seems that the authors took great care to rule out potential confounds (such as dropout artifacts and vividness), used both controlled and naturalistic stimuli, performed partial replication of their results, and showed that the effects are strong enough to appear in all individual subjects. The discussion is elegant and sheds light on the relevance of these findings to previous neuropsychological and cognitive results. Overall, I believe that the results of this paper are important to the field, as they combine several previous lines of research from the literature into a coherent framework, shed light on the possible basis of neurological impairments in scene perception and memory, and **may have implications for understanding the relation between perception and memory beyond just the spatial domain.***

My main comments below are about interpretation of the results in terms of distinction vs. overlap between the systems, and some missing discussion of relevant previous results from the literature.

We thank the reviewer for the careful and positive evaluation of our manuscript. We hope that our responses below adequately address the reviewer's comments, which have greatly strengthened the manuscript.

Major comments:

1. The authors stress at several points in the manuscript that the scene-perception and scene-memory systems are distinct from one another, and have a discussion paragraph related to the implications of this distinction/dissociation. While the results convincingly show that activity "migrates" anteriorly in all scene regions when transitioning from perception to memory, I am less convinced of the full dissociation between different systems, as the results may also indicate a combined system with gradual transition inside it. This is due to several reasons:

We appreciate the reviewer's deep consideration of our data and the nature of the overlap between the scene-perception and place-memory areas. We respond to each of the reviewer's points regarding the dissociation between the scene-perception and place-memory areas in line below.

a. The authors show a significant overlap between the regions identified for perception and memory (especially in the PPA and MPA), although this overlap was only assessed qualitatively. It could be a good idea to assess this overlap quantitatively, and test whether there is a sharp transition between the systems (perception-overlapping-memory) or a gradual transition.

We appreciate the suggestion to explore the nature of the transition between perception and memory areas (i.e. a gradual or sharp transition) in more detail. In response to the reviewer's comment, we analyzed the transition in place-memory activation in ventral temporal cortex (relative to PPA). We focused on ventral temporal cortex because, as the reviewer points out, the overlap in this area is complex, while the overlap on the lateral surface (clearly separable) and medial surface (clearly overlapping) are more straightforward.

First, in each participant, we divided the scene-perception area on the ventral surface (PPA) into 20 evenly spaced bins along its posterior-anterior axis and calculated the mean place-memory response (place vs people memory) of the surface vertices in each bin. These data are shown in Response figure 5; notably the y-axis shows demeaned and normalized place-memory responses, and therefore the direction of response (positive/negative) should not be interpreted. At the group level, in both the left and right hemisphere, the transition from perceptual to memory-related processing spanned approximately 50%-70% of the distance along the anterior-posterior axis of PPA. However, at the individual participant level, the transition was often sharper.

Response figure 5. Normalized place memory response along the posterior-anterior axis of the ventral scene-perception area (PPA). In each participant (light grey lines), PPA was divided into 20 evenly spaced bins, and we calculated the mean place-memory activity (place-memory versus people-memory) from each bin. The place-memory activity from each participant was then demeaned and normalized, and the group average was calculated (black line). At each bin, the red heatmap shows the proportion of participants with a place-memory area vertex within a given bin.

Together, these results suggest that the place-memory responsive region emerges rapidly from the anterior portion of the scene-perception area PPA, and that the transition from perception- to memory-responsive cortex has a sharp onset. However, these results are necessarily inconclusive, due to the limited spatial resolution of fMRI data. So, we have included this figure in the Extended data (Extended data figure 4) as a more direct quantification of the profile of place-memory related activity in ventral temporal cortex with this caveat.

Across the cortical surfaces, the degree of overlap between the scene-perception and place-memory areas varied systematically (Fig. 1, inset). On the ventral surface, place-memory activation emerged rapidly at the anterior 60-70% of the scene-perception area (PPA) and continued anteriorly beyond the scene-perception area's anterior extent (Extended data Fig. 4). In contrast, on the medial surface, the scene-perception area (MPA) was encompassed within the most posterior portion of the place-memory responsive region. On the lateral surface, the place-memory and scene-perception area (OPA) partially overlapped, but to a lesser extent than on the other cortical surfaces. This systematic variation across the cortical surfaces was present in all participants (Extended data figure 1a, Extended data file 1). (Page 3-4, Lines 94-102)

*b. The connectivity analysis in experiment 3 used partial correlation with other correlated regions, which can remove the common variance / signals between the memory and scene regions. **Therefore, the***

conclusion that the regions are not correlated in their activity across the networks seems too strong. In addition, while the differences in correlation of the two systems to the hippocampus and visual cortex appear striking, the difference of the between-network vs. within-network connectivity values appears to be driven only by the VMPPA-MPA weak correlation and not by other regions (at least it seems so visually from the correlation matrix).

1b i) We appreciate the opportunity to explain our analysis in Experiment 3 more clearly. We elected to use partial correlation to remove common variance across the regions of interest to control for the fact that all regions are receiving a strong, bottom-up sensory input from the naturalistic movie stimulus. This allowed us to test whether the scene-perception and place-memory areas separated into two distinct networks above and beyond their fluctuation about the common signal. As noted above, the signals from the scene-perception and place-memory regions are indeed highly correlated, most likely due to their collective visuo-spatial processing. Importantly, though, areas within the same network (MxM) are more correlated with each other than across networks (MxP). We have clarified this important point in the expanded results section of our manuscript:

We analyzed the co-fluctuation of activity among the scene-perception and place-memory areas during the movie by calculating the correlation between the activity timeseries for each region pair, whilst partialling out the activation of all other regions (Fig. 4a). Accounting for the activation of all other regions allowed us to assess the co-fluctuation of these areas above-and-beyond their shared response to the video stimulus, which causes all regions to exhibit overwhelming positively correlated activity. (Page 9, Lines 244-249)

1b ii) The reviewer raises an important question regarding whether outliers influence the results of Experiment 3. Specifically, the difference between within- and between-network correlated activity we observed could have been driven by the lowest between-network pairwise connections. To control for this possibility, we tested whether the difference was still significant after removing the connections between MPA-VPMA and PPA-LPMA (i.e. the two lowest pairwise connections in the matrix). Even when removing these connections, the difference between the within- and between-network correlated activity remained significant (SPNxSPN v PMN x SPN: $t(12) = 4.88$, $p = 0.0003$, $d = 3.11$; PMNxPMN v PMNxSPN: $t(12) = 3.27$, $p = 0.001$, $d = 2.39$). We have added this information to the expanded results section.

The difference between within- versus between-network correlations remained significant even after removing the two lowest pairwise correlation values (MPA with the ventral place memory area and PPA with the lateral place memory area; Perception:Perception v Perception:Memory, $t(12) = 4.88$, $p = 0.0003$, $d = 3.11$; Memory:Memory v Perception:Memory, SPN: $t(12) = 3.27$, $p = 0.001$, $d = 2.39$), which confirmed that the observed network-differentiation was not driven by outlying pairwise associations. (Page 9, Lines 256-261)

c. One important methodological issue is that the authors used in some of the analyses “constrained ROIs”. The methods describe “ROIs were constrained to unique members of the top 300 vertices from the perception and memory areas” – this is unclear – **I assume this means that the 300 voxels with the strongest contrast values from experiment 1 were taken in each region? This is a crucial point, as 1) it is not described in the main text and this may be misleading, and 2) this might result in getting non-overlapping ROIs from the originally overlapping ones. Therefore, the systems might seem artificially dissociated when they are actually partially overlapping: this is very relevant for interpretation of results in experiment 4 (memory regions not involved in perception and vice versa) and in experiment 3 (connectivity between and within systems). If my reading of the methods is correct, please describe this clearly and add it also to the main text.**

In my opinion, the issue of overlap does not affect the main conclusions of the paper or their importance, that would be equally valid if the systems are partially overlapping instead of distinct. But it is an issue that needs to be addressed and discussed, preferably with a formal measurement of the amount of overlap. If the systems are indeed partially overlapping, the analyses can also be performed on the parts of the two systems that are non-overlapping to show that parts of this system are dissociable while another part is common to both perception and memory.

1c) We thank the reviewer for the encouragement to clarify this important step in our methods (see also Reviewer 1, comments 1b and 2b). Yes, as the reviewer suggests, in Experiments 2-4, we considered only the top 300 unique and most scene-perception and place-memory selective surface vertices. The rationale behind this choice was that we wished to characterize and compare functionally homogenous regions (the top scene-perception-selective vs. place-memory-selective ROIs), rather than the large swaths of cortex that were significant at $p < 0.001$ (which likely overlapped with multiple anatomical areas and retinotopic maps). This also ensures that differences we see in activation or correlation were not due to differences in ROI size. We have updated the manuscript to make this procedure and our motivation clear to the reader:

To ensure that we evaluated functionally homogenous regions and to control for differences in ROI sizes across regions, we constrained the scene-perception and place-memory areas to the unique members of the top 300 most scene-perception/place-memory preferring vertices for all subsequent experiments. Importantly, on all cortical surfaces, the constrained place-memory area was anterior to the scene-perception area (Extended data Fig. 7). (Page 6-7, Lines 177-182)

The reviewer also expresses concern that this analysis step may present the systems as dissociable, rather than partially overlapping. We agree that this is an important point to clarify to the reader and have clarified this point in two ways. First, we added Extended data figure 4 (Response Figure 5) to better characterize the transition in scene vs. place selectivity across the extent of VTC, which illustrates the partially overlapping nature of these regions on the ventral surface. Second, we have revised the Discussion of our manuscript to clearly emphasize that these networks are partially overlapping in their

spatial extent, especially on the ventral and medial surfaces (Figure 1 and Extended data Fig. 1). Moreover, although we find that the peak 300 nodes of these networks are dissociable based on their functional profile, both during movie watching and also in memory-recall and recognition tasks, we acknowledge that both of the networks deal with spatially relevant information, and likely work in concert to facilitate visually guided navigation. We hope that extensive revisions to the Main Text and Discussion sections have addressed the reviewer's concern.

Across the cortical surfaces, the degree of overlap between the scene-perception and place-memory areas varied systematically (Fig. 1, inset). On the ventral surface, place-memory activation emerged rapidly at the anterior 60-70% of the scene-perception area (PPA) and continued anteriorly beyond the scene-perception area's anterior extent (Extended data Fig. 4). In contrast, on the medial surface, the scene-perception area (MPA) was encompassed within the most posterior portion of the place-memory responsive region. On the lateral surface, the place-memory and scene-perception area (OPA) partially overlapped, but to a lesser extent than on the other cortical surfaces. This systematic variation across the cortical surfaces was present in all participants (Extended data figure 1a, Extended data file 1). (Page 3-4, Lines 94-102)

However, intriguingly, the degree of overlap varied systematically by cortical surface. Specifically, on the ventral surface, the scene-selective area PPA was activated by scene perception and located immediately posterior to an area that was activated by both perception and memory (the ventral place memory area, VPMA) [...] The present results [...] suggest] that a distinct area might subserve this process: the place-memory responsive region is well-positioned to integrate visual information represented in PPA with the allocentric spatial representations thought to reside in hippocampus and entorhinal cortex^{11,62}. Thus, by introducing areas that may interface between visual cortex and memory structures, the present work fills this important gap in our understanding of how the brain may implement visually guided behaviors. (Page 13-14, Lines 358-380)

In contrast to the organization of the ventral surface, the scene-selective visual area on the medial surface, MPA, was activated by both scene perception and place memory and was contained within a larger place-memory responsive area (the medial place memory area, MPMA) that extended anteriorly into medial parietal cortex and retrosplenial cortex proper (Brodmann's area 29/30). [...] Thus, the present work unifies prior work by describing the topographical and functional distinction between MPA and MPMA: MPA may be best described as a visually responsive subregion of a larger, memory responsive area. [...] (Page 14, Lines 381-401)

On the lateral surface, scene-selective OPA was activated by perception and located immediately posterior to an area that was activated only during memory (the lateral place memory area, LPMA). Intersecting vertices between the two regions showed joint

responsivity to both perception and memory. Among all of the surfaces, the place memory area on the lateral surface is perhaps most topographically striking and theoretically surprising. [...] (Page 15, Lines 402-419)

2. The authors write in the abstract “we report a new network of brain areas... we reveal three cortical areas... This new network...” and so on, and in the introduction they write “We henceforth refer to these previously undescribed regions as ‘place-memory areas’”. These claims of “finding new areas and networks” seems exaggerated. Subdivisions of the PPA to posterior and anterior parts involved in perception and memory respectively have been suggested by Epstein 2008 (TICS) and Baldassano et al. 2013 (NeuroImage), who also showed that these subregions have differential connectivity patterns. Baldassano et al. 2016 (eNeuro) also showed differential connectivity in anterior and posterior PPA and OPA (but not MPA) and suggested that these constitute distinct networks for vision and memory. Findings of differences in processing in MPA posterior-anterior subdivisions were described in Silson et al. 2019 (J Neuroscience), Burles et al. 2017 (Brain Struct Func) and 2018 (Front Hum Neurosci), and two papers by Chrastil et al. 2018 (both in Behavioral Neuroscience). In the present paper the authors convincingly combine these lines of research into a coherent framework that spans all scene regions, show the different results in the same subjects and at the single-subject level, and extend the knowledge of the scene-selectivity and type of memory operations these regions are involved in. However, the claims of finding a “new network of previously undescribed brain areas” seems overblown, as the anterior regions and their involvement in scene memory have been investigated and discussed before. I would suggest removing these claims and changing them to something more neutral (e.g. “we report a network... we identified three cortical areas...”). In addition, I think that a thorough discussion of the previous papers described above (or other relevant ones) and how they relate to the current findings is important to add.

We thank the reviewer for appreciating the significance of our results, which we think tie together the disparate observations in the literature they have cited. We acknowledge the need to better contextualize our results. In the revised manuscript we have greatly expanded our discussion of these and other works (Page 13-15, Lines 358-419).

Minor comments / suggestions:

1. In the abstract, the authors write “This new network, which conveys memory signals to visually responsive posterior cortex” – as far as I could tell, the authors didn’t show that the anterior regions convey any signals to the visual regions, so this claim should be removed (although this could be a nice point for the discussion).

Agreed. We have removed this speculative point from the abstract.

2. In experiment 1, the authors show that in face memory and perception there is no similar shift between adjacent posterior-anterior regions, and have an interesting discussion paragraph dedicated to not finding a “face memory network”. However, looking at the individual subjects in the supplementary

gif file, it seems that there is a region that is more active for face memory than perception – in the medial parietal lobe (in line with some of the authors’ recent eLife paper). If this stands statistical analysis, it might be an interesting finding to briefly report or show in a supplementary figure (although it’s not the main subject of the paper).

We thank the reviewer for their careful reading of our data, and we agree that the finding of “no face memory network” should be made clearer in the context of our prior work (Silson, Steel et al 2019).

The reviewer is correct that we do observe an area in medial parietal cortex that preferentially activates during face- compared to place-memory in the present data, which is consistent with our previous work (Silson, Steel, et al. 2019) as well as others (e.g. Peer et al. 2016). Importantly, the place-memory area is anatomically adjacent/overlapping with an area of the scene-perception network and activates when viewing images of scenes (MPA), which is not the case with the people-memory area in medial parietal cortex: no portion of the face-memory responsive region on the medial surface is active when viewing images of faces. Likewise, we did not find strong evidence for a people-memory area in ventral temporal cortex outside of the fusiform face area. We have clarified this in the revised manuscript:

Finally, the distinct topography of the scene-perception and place-memory networks in posterior cerebral cortex was unique to the spatial domain – we found no separable networks for face-perception and memory (although, consistent with prior work³³, we did observe a people-memory preferring area in medial parietal cortex in most participants). (Page 17, Lines 495-498)

3. In experiment 2, the authors show that the scene memory regions responded more strongly to familiar vs. unfamiliar stimuli, and this was significantly stronger than in scene perception regions. But did scene perception regions by themselves also respond more strongly to familiar vs. unfamiliar stimuli? It seems so in Figure 2B. While this does not affect the conclusions, it is interesting in itself and can be easily tested.

In response to the reviewer’s query, we analyzed whether the scene-perception area responded preferentially when viewing familiar compared to unfamiliar panning movies. As the reviewer predicted, the scene-perception areas responded significantly more strongly when perceiving familiar compared to unfamiliar place videos. Importantly, the “familiarity response” was greater in the place-memory compared to scene-perception areas. We have added these details to the results section of the manuscript.

The scene-perception areas were also driven more by familiar compared to unfamiliar videos (OPA: $t(13) = 4.38$, $p = 0.0001$, $d = 1.171$; PPA: $t(13) = 4.46$, $p = 0.0001$, $d = 1.192$; MPA: $t(13) = 7.231$, $p < 0.0001$, $d = 1.932$) but to a significantly lower extent than the place-memory areas, as noted above. (Page 7, Lines 201-204)

4. The authors refer in experiment 1 to their memory task as “scene memory”, and seem to refer to the exact same task instructions as “scene imagery” in experiment 4. I found this confusing – if there is a difference, please state it clearly; if there isn’t, please use the same terminology to describe these results and tasks.

We thank the reviewer for raising this point, and upon consideration we think that the reviewer is correct. We have clarified the terminology in the manuscript to account for this, now referring to our task in Experiment 4 as scene “recall (i.e. mental imagery)” (e.g. line 295).

5. The authors discuss the scene memory regions as being part of the default-mode network, but they don’t actually show this anywhere besides them being in the same general parts of the brain. This overlap can be easily measured by using templates from the literature or by extracting the DMN from their own movie-watching data, although this can also be solved by saying something like “these seem to lie in the same parts of the cortex... future studies directly comparing these systems could help elucidate this issue”.

The reviewer raises an interesting point regarding the relationship between the default mode network and the place-memory areas, which we are excited to address in a future publication! For the reviewer, we have visualized the participant’s data relative to the default mode network (derived from Yeo et al. 2011) below (Response figure 6). However, given that a direct comparison of the place-memory network and default mode network is a direction we hope to tackle in future work, we have left the speculative discussion of the relationship between the default mode network and the place memory network based on the observations in our previous work (Silson, Steel et al. 2019, Figure 8a).

<<< Manuscript additions >>>

Response figure 6. The cortical co-localization of the scene-perception, place-memory, and default mode networks in an example subject. Scene-perception areas (white outlines) and place-memory (orange outlines) were defined for a single participant as described in Experiment 1. The default mode network was taken from the group parcellation published in Yeo et al. 2011. The place-memory network consistently resides between the scene-perception and place-memory network, suggesting that this area may bridge processing of visual and mnemonic information processed by these networks.

6. *General organization suggestions – I believe that readers will benefit from a summary of the main results of all of the experiments somewhere in the paper, as it is currently difficult to create a mental overview of the paper’s results without careful reading and note-taking through all of it. In addition, some of the figures – particularly figure 2 – might benefit from subheadings in the figure, or an in-figure legend of abbreviations (e.g. adding region abbreviations next to the appropriate cortical regions in Figure 2A), as the number of abbreviations used is slightly overwhelming and requires going back-and-forth to the legend to understand them.*

We thank the reviewer for this suggestion. In response, we have inserted a overview of all studies in the Introduction (Page 2-3, Lines 57-69) and an overview of the results in the Discussion (Page 13, Lines 337-348). We have also revised Figure 2 from our original submission to include subheadings, which are linked to the figure legend (now Figure 3 and 4, reproduced below).

Figure 3.

Figure 4.

7. Supplementary figure 1 has a very low resolution – it would be better if the resolution could be increased in the final version.

We thank the reviewer for bringing this to our attention. We have increased the resolution in the latest version of the manuscript

Reviewer #3 (Remarks to the Author):

Steel and colleagues report a series of 4 fMRI experiments in which they identified and characterized three cortical areas that contribute preferentially to place memory in comparison to scene perception. As demonstrated in a group of 14/13 participants, these areas are localized on the lateral, ventral and medial cortical surfaces, sit immediately anterior to brain regions that are preferentially involved in scene perception (i.e. the PPA, OPA, and MPA), and demonstrate greater functional connectivity with each other in comparison to scene perception regions, pointing towards a cortical network for place memory.

The interface between memory and perception has been the focus of increasing interest over the past 10 - 15 years, and this study is, therefore, both timely and interesting. The inclusion of multiple experiments as well as the general convergence of findings across these experiments is a strength, and overall, the paper is clearly written. My major concern, however, is that the design of the experiments and the consideration of the findings lack the necessary finesse/precision to shed light on the functional and theoretical importance of these 'place memory' areas. In other words, while this study reports an interesting finding in the localization of these areas, the interpretation of the reported activity and its conceptual significance, at least as the findings are currently reported, is not clear.

*1) The current study **somewhat glosses over the heterogeneity and complexity of 'place memory'**. For instance, memory age (childhood vs. adulthood) and type (e.g. semantic vs. episodic) have been suggested to impact the neural correlates underlying place memory (e.g. Moscovitch/Nadel and colleagues, Squire and colleagues) and yet these factors were not taken into account when the place stimuli lists were created for each participant (e.g. participants could recall a childhood kitchen and/or a current kitchen), or when the data were analyzed. **Is there any additional insight that the authors could provide into this issue**, for instance, by providing additional details on the average age of the familiar memories and whether there was significant variance across participants?*

We agree that the complex and multifaceted nature of “place memory” requires further discussion in our manuscript. We now discuss this issue on page 6 of the manuscript (Lines 160-162).

In addition, we sought to provide further insight into what influences place-memory area activation. Although we lack the data to provide insight into how the average age of familiar memories impacts neural representations, we conducted an analysis to test whether the activation of the place memory activity scaled with the richness of participant’s memory, using “vividness” as a proxy for the richness. Importantly, the vividness ratings were collected outside of the scanning session, and participants were instructed to rate how vividly they could remember the person/place at the time of survey completion. We tested whether the “vividness” of a memory covaried with the magnitude of the place-memory network response in Experiment 1 using a linear mixed effects model with Network (scene-perception/place-memory), stimulus type (person/place), hemisphere (left/right), and vividness as factors. We found no overall effect of vividness in any region (All main effect of Vividness $F < 0.76$, $p > 0.38$), nor did we find any interaction between vividness and stimulus type (All Vividness x Stimulus type interactions $F < 1.57$, $p > 0.31$).

Together, these results suggest that the richness with which a participant can recall a given memory does not impact the activation of the scene-perception and place-memory network on a given trial. Importantly, although others (Dijkstra et al. 2017) have found that activity in visual areas scales with vividness of imagery, these effects were seen on trial-by-trial fluctuations with imagery measured concurrently. Thus, our results are complementary, rather than contradictory, to this prior work. We have added this new analysis to the manuscript’s Extended Data and describe the results in the Main Text:

Importantly, face and place memories are complex and multifaceted, and factors such as vividness and memory age can influence the activity elicited by memory recall⁴⁴⁻⁴⁶. However, the difference between categories was not due to subjective differences in recall ability, as participants reported equal subjective vividness for place and face memory recall, although vividness of recollection was not related to activation magnitude in place-memory or scene-perception areas (Extended data Fig. 6). (Page 6, Lines 159-164)

Extended data Fig. 6. Participants reported no difference in vividness between imagery for people and places. Outside of the scanner, participants rated each personally familiar stimulus for vividness of visual imagery on a scale of 1-5, where 1 indicates no imagery possible and 5 indicates “as if you were seeing the <stimulus>.” There was no significant difference in vividness between people and place stimuli ($t(13) = 0.92$, $p = 0.373$), suggesting that the topological difference between place and people imagery activation was not due to differences of vividness between the stimulus categories. Additionally, we sought to rule out that the richness of recollection might impact the activity of the scene-perception and place-memory areas. So, we tested whether the “vividness” of a memory covaried with the magnitude of the place-memory network response on a given trial during Experiment 1 using a linear mixed effects model with Network (scene-perception/place-memory), stimulus type (person/place), hemisphere (left/right), and vividness as factors. We found no overall effect of vividness in any region (Main effect of Vividness -- Lateral: $F(1,3716) = 0.7647$, $p = 0.3819$; Ventral: $F(1,3716) = 0.1657$, $p = 0.684$; medial -- $F(1,3716) = 0.0001$, $p = 0.99$), nor did we find any interaction between vividness and stimulus type (Vividness x Stimulus type interaction -- lateral: $F(1,3716) = 0.4437$, $p = 0.5054$; ventral: $F(1,3716) = 1.0179$, $p = 0.3131$; medial: $F(1,3716) = 1.5674$, $p = 0.2107$) or vividness and network (Vividness x Network interaction -- lateral: $F(1,3716) = 0.001$, $p = 0.97$; ventral: $F(1,3716) = 0.525$, $p = 0.46$; medial: $F(1,3716) = 0.016$, $p = 0.90$).

2) Related to the above, further details on the types of familiar and unfamiliar scene stimuli that were used would be helpful for the interpretation of the data. For instance, were there similar categories/types and numbers of different categories/types of scenes for the familiar and unfamiliar stimuli (e.g. indoor vs. outdoor, cluttered vs. uncluttered scenes, etc) across the different experiments?

We thank the reviewer for the opportunity to clarify our methods. Our stimuli were counterbalanced across participants, such that one participant's familiar stimuli were used as unfamiliar for other participants, which would control for any stimulus-level differences. We have emphasized this point in the methods section.

To control for low-level variation in stimuli, unfamiliar videos were randomly sampled from other participants' familiar videos. (Page 22, Lines 656-659)

Unfamiliar places videos for the perception condition were not repeated from Experiment 2. (Page 23, Lines 697-698)

3) One interesting finding is the response of the hippocampus to the viewing of familiar vs. unfamiliar panning movies, and moreover, the strong correlation between place-memory area and hippocampal activity. These data point towards a role for the hippocampus in place memory and appear somewhat discrepant to studies that have implicated the hippocampus in both perception and memory (e.g. work of Maguire group in the context of scene construction, Lee/Barense in the context of spatial feature conjunction processing). **Indeed, there are some who would perhaps suggest that the hippocampus could function as an interface between scene perception and memory, and that this structure could subserve representations/processes that support both perception and memory.** Discussion of this issue is important - for example, do the authors consider the observed **lack of hippocampal involvement during perception** a function of their task design or support for the notion that the hippocampus contributes to memory only? Similarly, some have argued for a perceptual and mnemonic role for the perirhinal cortex for objects/faces, and this region did not feature in the face-related findings for Experiment 1.

We thank the reviewer for this opportunity to clarify an important theoretical point: our data do not suggest a lack of hippocampal involvement during perception. In Experiment 2, we found that the hippocampus is driven by perception of both familiar and unfamiliar place videos, but, like the place-memory areas, the hippocampus is driven more strongly by familiar stimuli (Supplementary figure 3a). Similarly, in Experiment 4, when we directly compared mental imagery and perception, the hippocampus (and left hippocampus in particular) was active during both imagery and perception. Thus, our data supports the idea that the hippocampus is involved in both perception and memory and perhaps could subserve representations/processes that support both perception and memory as the reviewer suggests.

Critically, the *preferential* engagement during memory conditions (familiar vs. unfamiliar places) suggests that hippocampal activity may be related to contextual/associative processing, for example

representing the visual information outside of the current field of view or situating the current field-of-view within broader spatial knowledge that is available for familiar stimuli. So, the preferential connectivity between the hippocampus and the place-memory network does not suggest that the hippocampus is not involved in perception, per se, but rather that its involvement in ongoing perception may be in service of representing (and encoding) contextual information relevant to ongoing perception.

In light of the reviewer's comments, we have made the following addition to the discussion section.

In addition to residing at the intersection of transmodal and perceptual cortex, the place-memory areas lie at the intersection between perceptual areas and memory structures in the brain, including the hippocampus and entorhinal cortex. Although traditionally the hippocampus has been associated with episodic memory^{45,86}, recently the hippocampus has been implicated in both perceptual and mnemonic tasks, including scene construction⁸⁷⁻⁸⁹. Consistent with this view, we found that the hippocampus, like the place-memory areas, responded preferentially to perceiving familiar stimuli compared to unfamiliar stimuli, suggesting that it has a role in visual recognition-related processes. Interestingly, unlike the place-memory areas, when we directly compared perception and recall in Experiment 4, the hippocampus responded more during perception compared to recall, which is qualitatively more similar to the scene-perception areas than the place-memory areas. Together, these experiments suggest that the hippocampus is involved in both perceptual and memory processes, but we cannot delineate what the hippocampus specifically contributes in each case. For example, during perception of unfamiliar stimuli, the hippocampus may be encoding relevant perceptual features, or comparing the present visual stimuli versus other internal representations, or both. Future work is necessary to disentangle the role of the hippocampus in scene perception and situate it in the context of the place-memory and scene-perception networks. (Page 16, Lines 443-460)

*4) I would appreciate a little more detail and discussion regarding the actual **location (in terms of mean group max coordinates, anatomical structures)** of the different memory place areas. For example, **does the VPMA extend into parahippocampal cortex and more anteriorly into entorhinal cortex**, areas that have been implicated in spatial/contextual memory and navigation? Similarly, **how does the MPMA relate to the retrosplenial regions that have been implicated in navigation**, and episodic/autobiographical memory? Indeed, thinking more broadly, how do the authors view their place memory area network in relation to the network of regions that is typically associated with spatial navigation and spatial memory?*

We agree that readers will benefit from seeing the location of the place-memory areas contextualized with reference to accepted anatomical parcels, and we thank the reviewer for this suggestion. In

response, we conducted a group-analysis of the place-memory localizer and compared the results to two well-established parcellations: 1) a probabilistic retinotopic atlas (Wang et al. 2014), and 2) the anatomical parcellation from Glasser et al. 2016. This analysis yielded a striking pattern of results. First, the place-memory activity fell anterior to the cortical retinotopic maps, suggesting that, unlike the scene-perception areas, these areas do have overt visual field preferences. Second, when we compared the place-memory activity to the Glasser atlas, we found that the peak of place-memory activity fell at the boundaries between parcels associated with perception and spatial memory (Response figure 7). Specifically, in both the left and right hemisphere, the peak activation place memory area on each cortical surface was: MPMA -- parietaloccipital sulcus area 1; VMPPA -- parahippocampal area 1 and presubiculum; and LPMA -- area PGp.

Response figure 7. Group-based localization of place-memory preferring areas. Hot colors show the areas showing significant place-memory preferring responses, and outlined regions show parcels identified from a consensus multimodal parcellation (Glasser et al. 2016). Group data is thresholded at vertex-wise $p = 1e-6$, $q = 0.00015$.

We have included these analyses in the Main Text and added a new Figure (Figure 2) to show the group-analysis results in the left hemisphere relative to both retinotopic and anatomical parcellations.

Main Text addition:

We performed two additional analyses to characterize the cortical locations of the place memory areas. First, we asked: do the locations of the place-memory areas fall in retinotopic, or non-retinotopic cortex? We hypothesized that, unlike scene perception areas, which process visual input, place memory areas would lie in non-retinotopic cortex. To test this, we performed a group analysis of the place memory localizer and compared the peak activity for each area to retinotopic parcels based on a probabilistic atlas⁴². Strikingly, on all cortical surfaces, place-memory areas were consistently located anterior to the cortical retinotopic maps (Figure 2a) with minimal overlap (Extended data Fig. 5). This location in non-retinotopic cortex is potentially consistent with their role as preferentially mnemonic, rather than visual, areas. Second, we asked: how do the locations of the place-memory areas relate to known anatomical landmarks in the brain? To address this question, we compared the results of the group place-memory localizer to anatomical parcels from the Glasser atlas⁴³. This revealed a remarkable pattern: the peaks of the place-memory preferring activity fell at the intersection of areas known to be important for visual and spatial processing. Medially, the peak fell within the parietal occipital sulcus area 1 near the retrosplenial cortex; ventrally, the peak fell between parahippocampal area 1 and the presubiculum, and laterally the peak fell within area PGp and PGi (Figure 2b). Thus, the place-memory areas straddle visual and spatio-mnemonic functional areas, and, based on this localization, these areas are poised to play a role in bridging these processes. (Page 5, Lines 124-141)

Fig. 2. Location of place-memory activity in relation to known functional and anatomical landmarks. a. The place-memory areas are anterior to and have minimal overlap with retinotopic cortex. To understand the relationship between place-memory activity and known brain areas, we conducted a group analysis of the place-memory localizer (place > people memory recall; thresholded at vertex-wise $t > 7.3$, $p = 1e-6$) and compared the activation to the most probable location of the cortical retinotopic maps using the atlas (overlaid) from Wang et al. 2014. The peak of place memory activity was anterior to the retinotopic maps on the lateral and ventral surfaces and there was very little overlap. Between place-memory activity and these maps. **b.** The place-memory areas fall at the intersection between anatomical parcels known to be involved in visual processing and those associated with spatial memory. To understand the anatomical localization of the place-memory areas, we compared the peaks of place-memory activity with parcels from Glasser et al. 2016 (overlaid). We found that the place-memory areas fell at the intersection of parcels associated with visual and spatial processing. Activation maps are replotted in panels a and b to allow comparison between parcellations. To aid visual inspection only the left hemisphere is shown

5) The authors state that their findings are inconsistent with previous fMRI work that has suggested overlapping neural substrates for perception and mental imagery of high-level stimuli. According to my knowledge, however, in many previous fMRI studies on perception vs. imagery, participants were asked to perceive and visualize the same stimuli. This contrasts to the current study in which participants perceived unfamiliar places but imagined personally familiar places. **Thus, not only was there a difference in memory (unfamiliar vs. familiar) but also stimulus content, and it is unclear which of these is underlying the observed discrepancy with previous work.** I think the evidence (in particular Expt 2, Extended data Fig 7) suggests that memory is the key factor here but **perhaps the most convincing way to demonstrate this empirically would be to scan participants while they perceive unfamiliar**

scenes, allow them to then become highly familiar with the same scenes, and then scan them again when they are asked to imagine or see the same scenes again - an anterior shift in activity would provide further weight to the authors conclusions.

We thank the reviewer for deeply considering Experiment 4 of our study, and for the interesting question and suggested future experiment. The reviewer is correct that in many previous fMRI studies on perception v. imagery, participants perceive and visualize the same (familiar) stimuli (e.g. O’Craven and Kanwisher, 2002; Boccia et al. 2019). In other studies, however, participants perceive and visualize different stimuli (familiar/unfamiliar, respectively) (e.g. Ishai et al., 2002). Both approaches have consistently resulted in claims of an overlapping neural basis of perception and imagery. For example, Ishai et al. (2002), used a stimulus design similar to the design we employed -- participants either viewed unfamiliar stimuli (e.g. a house) or were told to imagine a house “from long-term memory”. Based on their results, they argued that memory and perception shared neural substrates. Thus, we do not believe that the choice of stimuli for imagery underlies the observed discrepancy with previous work. We now include this important point in our Discussion section:

Based on previous fMRI studies, it has been widely assumed that perception and recall (i.e. mental imagery) of high-level stimuli (such as scenes) recruit the same neural substrates, including category-selective areas in ventral temporal cortex^{31,90}. For example, prior studies found that both perceiving and recalling scenes evoked activity in overlapping, scene-selective cortical areas and concluded that perception and memory shared neural substrates^{22,29,47}. (Page 16, Lines 547-551)

Importantly, as in some previous studies of mental imagery⁴⁷, participants in our study recalled *familiar* locations in the recall condition but viewed *unfamiliar* locations in the perception condition. Future work should consider mapping the evolution of place-memory area response longitudinally as stimulus familiarity develops to better understand how familiarity shapes the activity of the place-memory areas. (Page 17, Lines 574-578)

6) Page 5 states that in Experiment 3 “participants watched a movie designed to concurrently engage perceptual and mnemonic representations of complex real-world environments”. I may have misunderstood something but this statement confused me since the Methods states that for the naturalistic movie, “Participants were not asked whether places were familiar to them”. Given this latter statement, how can the authors be certain that the naturalistic movie engaged both perceptual and mnemonic representations?

We thank the reviewer for pointing out this aspect of the manuscript that needs clarification. This statement in the methods refers to our post scan debriefing, at which point we did not ask participants whether they were familiar with the locations depicted in the video (e.g. whether the Dartmouth students we scanned were familiar with Cornell University’s campus). However, after the scan we did confirm that all participants did pay attention to the movie. Therefore, we assumed that participants

engaged short-term memory processes (memory built up over the course of the movie) by virtue of playing attention to the stimulus over time. In addition, we assumed that participants engaged long-term memory processes during segments of the video that included Dartmouth College, because our participants were Dartmouth students and employees. We have added this information to the Main Text of our manuscript to clarify this point for the reader.

Participants watched an 11-minute video that featured a concatenated series of architectural tours, college admissions videos (including Dartmouth College), and real estate advertisements (see Methods). This video was designed to concurrently engage perceptual and mnemonic representations of complex real-world environments by engaging short-term memory built up over the course of the video, and long-term memory processes engaged during video segments that included familiar locations (i.e. Dartmouth College). (Page 8-9, Lines 324-329)

7) Apologies if I missed this but I assume that the ROIs used for Experiment 3 were mutually exclusive in their extent (i.e. the neighbouring memory and perception ROIs that were used to calculate correlations in activity were not overlapping).

Yes, the reviewer is correct. The ROIs in Experiments 2-4 were mutually exclusive in their extent (only contained unique surface vertices). We have now clarified this important point in the Main Text of the manuscript.

To ensure that we evaluated functionally homogenous regions and to control for differences in ROI sizes across regions, we constrained the scene-perception and place-memory areas to the unique members of the top 300 most scene-perception/place-memory preferring vertices for all subsequent experiments. (Page 6-7, Lines 177-182)

8) Were the same stimuli repeated across experiments 1, 2, and 4? For instance, were the same unfamiliar panning movies used in Experiment 2 as in Experiment 4 (in which case, they may not have been as unfamiliar in Experiment 4 given the same participants would have seen them before?).

We thank the reviewer for bringing this point of clarification to our attention. No, the unfamiliar videos in Experiment 2 and 4 were non-overlapping: no unfamiliar videos were repeated across the Experiments. Thus, all videos were unfamiliar in both Experiments. In contrast, the familiar stimuli used in our recall/mental imagery tasks (Experiments 1 and 4) were drawn from the same list of personally familiar place stimuli. We have added these details to the methods section describing Experiment 4:

As in Experiment 3, 24 unique stimuli were used in each condition (24 place word stimuli taken from the list generated in Experiment 1, and 24 panning videos of unfamiliar places). (Page 23, Lines 692-694)

Unfamiliar places videos for the perception condition were not repeated from Experiment 2. (Page 23, Lines 697-698)

9) *A very minor point (and perhaps I just missed this) but it would be useful to describe the collection of the data for Extended data Figure 2 in the Methods section of the main manuscript.*

We have expanded our explanation of this data in the methods section of our manuscript.

To ensure that participants could reliably visualize the familiar stimuli used in Experiments 1 and 4, and to confirm that there were no differences in visualization between stimulus categories, we had participants rate their ability to richly visualize each stimulus. Specifically, after the scanning session we gave the participants the following instructions:

Please rate each stimulus on how well you are able to visualize each stimulus from 1-5 (1 – I cannot visualize this person/place, 5 – as though I am seeing the person/place). Please rate the stimuli base upon how well you can visualize the stimulus, and not try to remember how well you visualized the stimulus during the scan. (Page 20, Lines 598-606)

Reviewers' Comments:

Reviewer #1:

Remarks to the Author:

In this revised manuscript Steel and colleague made substantial revisions which improved the manuscript. While they provided detailed and thoughtful revisions and addressed many of my comments, they haven't addressed my main concerns, which I will reiterate below.

My main concern was that their definition of the place areas (PPA, OPA, MPA) was based on a contrast of places vs. faces, which is not a suitable contrast. All their subsequent results hinge on this definition. The reason that this is not a suitable contrast is because comparing only 2 categories identifies any voxels that respond to more to one category than the other but does not preclude the possibility that a 3rd or 4th or nth category has a higher response than either of these 2 categories. Thus, this is an insufficient contrast to define category selectivity. In order to define a category-selective region the standard in the field is contrasting the response to one category vs a variety of other categories. Put another way, the place areas marked and analyzed in this paper are inconsistent with the standard in the field and this needs to be corrected prior to publication. Any conclusion drawn from the present ROI analyses is not sufficient to provide support for the authors' claims.

Main concerns:

In their response the authors (i) compared places to faces and objects and put an image showing this contrast in extended fig 3 , (ii) compared their place memory to additional atlases (fig 2 and extended fig 3,5) and (iii) reported that their ROI analyses are actually not based on ROIs shown but actually on the 300 top voxels for the respective task

This response is insufficient for several reasons.

First, showing that the map in figure 1 and extended figure 3 are overlapping is insufficient to address my concerns. To address my concern, the authors would need to replace figure 1 and extended figure 1 using the contrast places vs faces + objects and redo all the subsequent analyses related to experiments 2-4 using this definition of place vs faces & objects. They could put a control analysis with the present figure 1 and extended figure 1 as extended figures.

Second, a related analysis that I suggested in the original review and was not done in the revision, was to do an ROI analysis of data in experiments 2-4 using an independent ROI from a different study that used a contrast for place-selectivity using multiple categories such as the Weiner 2018 PPA (or the Wang PHC1/2, which is based on retinotopic criteria but overlaps the PPA). This would (i) use an independent ROI that is validated and (ii) control for ROI size across participants. This independent analysis is important for verifying that the place-perception area does not show enhanced responses to place memory or imagery or familiarity in subsequent experiments or have differential functional connectivity.

A third concern is that their response highlighted that the ROIs shown in fig 1 and extended figure 1 are actually not the ROIs used for subsequent ROI analyses. In fact, they used the 300 top voxels within each ROI. This is concerning for 2 main reasons. First, reporting should be transparent. The ROIs depicted in the figure should match the data shown from these ROIs. Presently, the figures show different ROIs than the ones actually analyzed. So, at minimum, the ROIs in the figure should depict the ROIs from which the data are taken. Second, choosing the top 300 voxels does not constrain the voxels to be clustered spatially, and in fact they can be randomly spread across the depicted territory. Given that the authors make claims about place-perception areas and place-memory areas it is pertinent to their hypothesis that ROIs are spatially clustered. So, their data analyses methods should be consistent with their theoretical questions. A different approach to control size would be to restrict the ROI to a disk around the center of the ROI. They use the centers to show the shift from perception to memory, the centers tend to be stable across runs and days, and using a disk would produce a clustered ROI of constant size. A more minor comment is that they should justify why they chose a

number of 300 voxels. I understand that thresholds are somewhat arbitrary, but criteria need to be justified.

So, in the revision they should use the place > face + object contrast and if they want to further constrain the ROIs, they should illustrate these constrained ROIs in the figures. Btw, the same comment holds for the definition of the face area.

Finally, I really liked the new figure 2, which anchors their findings compared to other prevalent atlases used in the field. However, I have 2 concerns regarding this new figure: (i) they use a much higher threshold here $t > 7.3$ compared to $t > 3.3$ in the other figures. I see no reason to change the thresholds across figures; So please update all figures to use the same threshold. (ii) Please show in additional panels the place-perception contrast from your study compared to the Glasser and Wang atlases, again using the same threshold.

Reviewer #2:

Remarks to the Author:

The authors have adequately addressed all of my comments. This is an excellent paper and I recommend publication.

(One small comment - in line with my previous remarks, I would remove the words "previously undescribed" (line 64)).

Reviewer #3:

Remarks to the Author:

The authors have done an excellent job dealing with the concerns I had raised in my previous review. From my perspective, the findings will be an interesting and potentially important addition to the literature.

1 We thank the editor for considering our manuscript for a second revision, and for the enthusiastic
2 assessment of our manuscript by Reviewers 2 and 3.

3
4 In response to Reviewer 1, we have conducted additional analyses to address their remaining concerns.
5 All additional analyses confirm our results as originally reported. Specifically, we have:

- 6
7 • Reanalyzed Experiments 2-4 using the Reviewer's suggested definition for the scene perception
8 area region of interest (Scenes > Faces + Objects). In all experiments, our results were
9 unchanged.
- 10 • Reanalyzed Experiments 2-4 using a disk ROI, as the reviewer suggests. As above, all results
11 were unchanged.

12
13 We thank the reviewer for recommending these analyses, which further reinforce the robustness of our
14 findings.

15
16 We also note that we have added two words to our title to more precisely describe our results: "A
17 network linking scene perception and spatial memory systems in the human brain."

18
19 Our responses are detailed in-line below. The comments from Reviewer 1 are in italics, our responses
20 are in blue, and additions to the manuscript are in red.

21
22 We hope that you find our manuscript suitable for publication in *Nature Communications*.

23
24 Sincerely,

25
26 Adam Steel

27 *On behalf of the authors*
28

29
30
31 *Reviewer #1 (Remarks to the Author):*

32
33 *In this revised manuscript Steel and colleague made substantial revisions which improved the*
34 *manuscript. While they provided detailed and thoughtful revisions and addressed many of my*
35 *comments, they haven't addressed my main concerns, which I will reiterate below.*

36
37 *My main concern was that their definition of the place areas (PPA, OPA, MPA) was based on a contrast*
38 *of places vs. faces, which is not a suitable contrast. All their subsequent results hinge on this definition.*
39 *The reason that this is not a suitable contrast is because comparing only 2 categories identifies any*
40 *voxels that respond to more to one category than the other but does not preclude the possibility that a*
41 *3rd or 4th or nth category has a higher response than either of these 2 categories. Thus, this is an*
42 *insufficient contrast to define category selectivity. In order to define a category-selective region the*
43 *standard in the field is contrasting the response to one category vs a variety of other categories. Put*
44 *another way, the place areas marked and analyzed in this paper are inconsistent with the standard in the*
45 *field and this needs to be corrected prior to publication. Any conclusion drawn from the present ROI*
46 *analyses is not sufficient to provide support for the authors' claims.*
47

48 *Main concerns: In their response the authors (i) compared places to faces and objects and put an image*
49 *showing this contrast in extended fig 3 , (ii) compared their place memory to additional atlases (fig 2 and*
50 *extended fig 3,5) and (iii) reported that their ROI analyses are actually not based on ROIs shown but*
51 *actually on the 300 top voxels for the respective task.*

52

53 We thank the reviewer for clearly outlining their concerns, which we address in turn below.

54

55 1) *This response is insufficient for several reasons. First, showing that the map in figure 1 and extended*
56 *figure 3 are overlapping is insufficient to address my concerns. To address my concern, the authors*
57 *would need to replace figure 1 and extended figure 1 using the contrast places vs faces + objects and*
58 *redo all the subsequent analyses related to experiments 2-4 using this definition of place vs faces &*
59 *objects. They could put a control analysis with the present figure 1 and extended figure 1 as extended*
60 *figures.*

61

62 In response to the reviewer's comments, we have rerun all analyses from Experiments 2 through 4
63 based on the contrast of scenes > faces + objects as suggested. **We find our results are completely**
64 **unchanged.** Specifically, when we used the reviewer's suggested definition, we found that:

65

- 66 • (Experiment 2) The place memory areas exhibit an enhanced familiarity effect when viewing
67 panning videos of scenes compared to the scene perception areas.
- 68 • (Experiment 3) The scene perception and place memory areas form distinct functional networks
69 and preferentially associate with the primary visual cortex and hippocampus, respectively.
- 70 • (Experiment 4) The scene perception and place memory areas have dissociable activity when
71 perceiving and recalling scenes.

72

73 These results are detailed below, with figures and the corresponding statistics in the figure legends. We
74 also note that these unchanged results are unsurprising: as shown in the statistical analyses provided in
75 Revision 1, which the reviewer does not reference in their comments here, the location and extent of
76 scene-selective regions did not statistically differ between the two contrasts under debate ("scenes >
77 faces + objects" vs. "scenes > faces").

78

79

80
 81 **Response figure 1. Experiment 2: Place memory areas exhibit an increased response to viewing**
 82 **panning movies of familiar places that is greater than the scene perception areas defined using the**
 83 **contrast Scenes > Faces + Objects. Lateral (LPMA v OPA): $t(13) = 2.65$, $p = 0.02$, $D = 0.7$; Medial (MPMA**
 84 **v MPA): $t(13) = 5.15$, $p = 0.00019$, $D = 1.13$; Ventral (VPMA v PPA): $t(13) = 3.18$, $p = 0.007$, $D = 0.85$**

85
 86
 87
 88

89
 90 **Response figure 2. Experiment 3. (left) The place memory areas and scene perception areas exhibit**
 91 **greater within network connectivity compared to between network connectivity when the scene**
 92 **perception areas defined using the contrast Scenes > Faces + Objects. P v PxM: $t(12) = 2.97$, $p = 0.01$, D**
 93 **$= 2.11$; M v PxM: $t(12) = 2.20$, $p = 0.048$, $D = 1.59$; P v M: $T(12) = 3.4$, $p = 0.005$, $D = 0.93$. (right) The**
 94 **scene perception network is more connected to primary visual cortex than the place memory**
 95 **network, while the place memory network is more connected to hippocampus than the scene**
 96 **perception network when the scene perception areas are defined using the contrast Scenes > Faces +**
 97 **Objects. PxV1 - PxHipp v MxV1 - MxHipp: $t(12) = 6.3$, $p < 0.0001$, $D = 4.52$. This confirms the results**

98 presented in the main text where the scene perception areas were defined using the contrast Scenes >
 99 Faces.
 100
 101

102 **Response figure 3. Place memory areas respond more when recalling familiar places compared to the**
 103 **scene perception areas when defined using the contrast Scenes > Faces + Objects.** Lateral: Perception -
 104 $t(13) = 7.092$; $p < 0.0001$, $D = 2.3$, Memory - $t(13) = 8.509$; $p < 0.0001$, $D = 3.94$; Ventral: Perception -
 105 $t(13) = 4.803$; $p = 0.0003$, $D = 1.19$, Memory - $T(13) = 6.50$; $p < 0.0001$, $D = 2.77$; Medial: Perception -
 106 $t(13) = 0.83$; $p = 0.41$, $D = 0.25$, Memory - $T(13) = 4.82$; $p = 0.0003$, $D = 1.17$. This confirms the results
 107 presented in the main text where the scene perception areas were defined using the contrast Scenes >
 108 Faces.
 109

110
 111
 112
 113 The reviewer also suggested that we revise all figures and results in our Main Text to use their preferred
 114 contrast for defining scene-selective regions. We respectfully disagree with this request for the following
 115 reasons:

- 116 • First, our results are not influenced by the choice of ROI definition: neither the location of scene-
 117 selective regions (Revision 1) nor the statistical results of any of our experiments (Revision 2) are
 118 affected.
- 119 • Second, although the results are equivalent regardless of contrast choice, the scenes > faces
 120 contrast is most analogous to the contrast used to define the memory areas. The comparison
 121 between perception and memory areas is central to the paper, and it is therefore preferred that
 122 we match these conditions as best as possible.
- 123 • Third, we respectfully disagree with the reviewer’s claim that our preferred contrast is non-
 124 standard in the literature: contrasting activation in response to scene versus face images is a
 125 widely accepted definition of the scene-selective areas. We have used this definition in our past
 126

work (e.g., Silson et al. 2015, *JNeuro*; Silson, Steel, and Baker 2016, *Frontiers in Neuroscience*; Robertson et al. 2016, *Current Biology*; Silson et al. 2019, *JNeuro*) and has been used extensively by others investigating perception and memory functions of these areas (Coggan, Baker, and Andrews, 2018, *EJN*; Henriksson, Mur, and Kriegeskorte 2019, *Neuron*; Boccia et al. 2017, *Brain structure and function*; Boccia et al. 2019, *Human brain mapping*; Groen et al. 2017, *eLife*; King et al. 2018, *NeuroImage*, Kravitz et al. 2013, *JNeuro*; Berens et al., 2020, *J. CogNeuro*; among many others).

We have now included a sentence in the Methods section: “Importantly, our findings in Experiments 1-4 remained significant when two additional approaches to defining scene-selective areas were used: 1) the contrast scenes > faces + objects, and 2) a disk of 300 nodes drawn around the center of each ROI (See Extended Data Figs. 14-15) (Page 26)”.

2) Second, a related analysis that I suggested in the original review and was not done in the revision, was to do an ROI analysis of data in experiments 2-4 using an independent ROI from a different study that used a contrast for place-selectivity using multiple categories such as the Weiner 2018 PPA (or the Wang PHC1/2, which is based on retinotopic criteria but overlaps the PPA). This would (i) use an independent ROI that is validated and (ii) control for ROI size across participants. This independent analysis is important for verifying that the place-perception area does not show enhanced responses to place memory or imagery or familiarity in subsequent experiments or have differential functional connectivity.

The reviewer raises the concern that the ROIs used in our experiment are 1) not sufficiently independent or validated and 2) may differ in size from each other. They suggest that we use the ROI from either Weiner 2018 or Wang 2014 to define PPA to address this concern. We disagree with these concerns and the accompanying suggestion for three reasons.

i) The ROIs used in our study for Experiments 2-4 were independently defined in each participant using data from Experiment 1. Therefore, our analyses are independent (i.e., not circular).

ii) Our ROIs are controlled for size. We shared the reviewer’s concern regarding controlling for ROI sizes (both between memory and perception areas, and across people). To address this, we adopted an approach that is commonly used in the field that we have used in our past work (Silson, Steel et al. 2020) and has been used by others (e.g., Julian, Ryan, and Epstein 2017, *Cerebral Cortex*; Bonner and Epstein 2017, *PNAS*; Bonner and Epstein 2018, *PLOS Computational Biology*; Marchette et al. *JNeuro*, 2015; Julian et al. *NeuroImage* 2012): to perform statistical analyses based on the top X voxels of any region, where X is held constant across regions and participants. Thus, we controlled for ROI size across participants. Please see also our response to Comment #3 below, where we show that our results hold when using another commonly accepted approach for controlling for ROI size (disks centered on each ROI).

iii) The suggestion to use ROIs from Weiner 2018 or Wang 2014 to define PPA is not desirable or even possible for three reasons. First, ROIs from Weiner 2018 or Wang 2014 would only allow us to define the scene-perception area on the ventral surface, PPA, but our paper considers all three regions of the scene network. Using our functional definitions of the scene perception areas ensures that our regions are defined consistently across all cortical surfaces. Second, a key strength of our paper is that all ROIs are defined within-subject, allowing us to produce a fine-

175 grain comparison between the location of mnemonic and perceptual areas. Adopting a group-
176 defined probabilistic atlas ROI goes against this approach. Third, in Revision 1, we showed that
177 our functional localizer successfully identifies the most probable location of PPA, as defined by
178 Weiner 2018 (Extended data Figure 3) and Wang 2014 (Extended data Figure 5). This suggests
179 that the results from Experiments 2-4 would be unchanged if we used these probabilistic ROI
180 definitions.

181
182
183 3) A third concern is that their response highlighted that the ROIs shown in fig 1 and extended figure 1
184 are actually not the ROIs used for subsequent ROI analyses. In fact, they used the 300 top voxels within
185 each ROI. This is concerning for 2 main reasons. First, reporting should be transparent. The ROIs depicted
186 in the figure should match the data shown from these ROIs. Presently, the figures show different ROIs
187 than the ones actually analyzed. So, at minimum, the ROIs in the figure should depict the ROIs from
188 which the data are taken. Second, choosing the top 300 voxels does not constrain the voxels to be
189 clustered spatially, and in fact they can be randomly spread across the depicted territory. Given that the
190 authors make claims about place-perception areas and place-memory areas it is pertinent to their
191 hypothesis that ROIs are spatially clustered. So, their data analyses methods should be consistent with
192 their theoretical questions. A different approach to control size would be to restrict the ROI to a disk
193 around the center of the ROI. They use the centers to show the shift from perception to memory, the
194 centers tend to be stable across runs and days, and using a disk would produce a clustered ROI of
195 constant size. A more minor comment is that they should justify why they chose a number of 300 voxels. I
196 understand that thresholds are somewhat arbitrary, but criteria need to be justified.

197
198 So, in the revision they should use the place > face + object contrast and if they want to further constrain
199 the ROIs, they should illustrate these constrained ROIs in the figures. Btw, the same comment holds for
200 the definition of the face area.

201
202 The reviewer is concerned that our choice to constrain our regions of interest to the 300 most scene
203 selective surface vertices may cause our regions to be non-contiguous. They suggest using a disk,
204 centered on the peak of selectivity for perception (scenes > places) and memory (places > people), as an
205 alternative means to define these regions. Following this suggestion, **we have rerun all analyses**
206 **associated with Experiments 2-4, and we find that our results are unchanged.** The critical comparisons
207 for each experiment and associated figures are shown below.

208
209

210
 211 **Response figure 4. Experiment 2. Place memory areas exhibit an increased response to viewing**
 212 **panning movies of familiar places that is greater than the scene perception areas are defined using**
 213 **the 100 vertices closest to the localizer activation peak. Medial: $t(13) = 5.98$, $p < 0.0001$, $D = 1.60$;**
 214 **Ventral: $t(13) = 4.80$, $p = 0.0003$, $D = 1.28$; Lateral: $t(13) = 4.32$, $p = 0.0008$, $D = 1.28$.**
 215
 216

217
 218 **Response figure 5. Experiment 3. (left) The place memory areas and scene perception areas exhibit**
 219 **greater within network connectivity compared to between network connectivity when the scene**
 220 **perception areas defined using the 100 vertices closest to the localizer activation peak. P v PxM: $t(12)$**
 221 **= 4.65, $p = 0.0005$, $D = 2.84$; M v PxM: $t(12) = 3.45$, $p = 0.004$, $D = 2.14$; P v M: $t(12) = 0.23$, $p = 0.79$, $D =$**
 222 **0.1. (Right) The scene perception network is more connected to primary visual cortex than the place**
 223 **memory network, while the place memory network is more connected to hippocampus than the**
 224 **scene perception network when the scene perception areas are defined using the contrast Scenes >**
 225 **Faces + Objects. PxV1 - PxHipp v MxV1 - MxHipp: $t(12) = 5.8$, $p < 0.0001$, $D = 4.14$.**
 226

227
 228 **Response figure 5. Experiment 4. Place memory areas respond more when recalling familiar places**
 229 **compared to the scene perception areas when the areas are defined using the 100 vertices closest to**
 230 **the localizer activation peak.** Lateral: Perception – $t(13) = 6.91$, $p < 0.0001$, $D = 2.43$, Memory – $t(13) =$
 231 7.53 , $p < 0.0001$, $D = 2.7$; Ventral: Perception – $t(13) = 3.83$, $p = 0.002$, $D = 1.15$, Memory – $t(13) = 6.34$, p
 232 < 0.0001 , $D = 1.85$; Medial: Perception – $t(13) = 5.7$, $p < 0.0001$, $D = 1.1$; Memory – $t(13) = 3.97$, $p =$
 233 0.0015 , $D = 1.12$.

234
 235
 236 Importantly, however, we disagree with the reviewer that our method for ROI definition is non-
 237 standard. Constraining to a specific number of nodes in an automated way is a widely accepted practice
 238 that we have used in our past work (Silson, Steel et al. 2020) and has been used by others (e.g., Julian,
 239 Ryan, and Epstein 2017, *Cerebral Cortex*; Bonner and Epstein 2017, *PNAS*; Bonner and Epstein 2018,
 240 *PLOS Computational Biology*; Marchette et al. *JNeuro*, 2015; Julian et al. *NeuroImage* 2012; Murty et al.
 241 2020, *PNAS*; Isik et al. 2020, *PNAS*). As detailed in response to Comment 2 above, this method controls
 242 for potential differences in ROI sizes between regions and individuals, ensuring that the regions of
 243 interest have similar numbers of vertices, and are selected in an unbiased manner.

244
 245 In addition, the reviewer expresses concern that we have not been transparent in reporting the
 246 thresholding procedure. However, we note have clearly and explicitly mentioned this procedure in the
 247 following locations:

- 248 • Main text: Page 6, Line 178-179
- 249 • Methods: Page 25, Line 781
- 250 • Extended data Figure 7.

251
 252 In the interest of being as transparent as possible, we have revised Extended data Figure 7 to show the
 253 constrained region of interest in our example participant (see below) in addition to showing the anterior
 254 shift of the top 300 memory-selective vertices compared to the scene-perception vertices.

255
 256
 257
 258

**Top 300 nodes
Subj001**

259
260
261
262
263
264
265
266
267
268
269
270
271
272
273
274
275

Extended data Figure 9. **Anterior shift of place-memory relative to scene perception remains consistent after constraining to maximally selective surface vertices.** (Left). Top 300 in each participant most scene-selective surface vertices were used for as regions of interest for Experiments 2-4. One example participant (Subj001, used in Figure 1) is shown. (Right). Across the lateral, medial and ventral surfaces, we compared the weighted center-of-mass of both the scene-perception and place-memory selective areas after constraining the regions to the top 300 most scene-perception/place-memory preferring surface vertices. Using the constrained ROI definitions, we still observed a significant anterior shift from scene-perception to place-memory on all cortical surfaces in both the left and right hemisphere (Lateral - left hemisphere: $t(13) = 9.35$, $p < 0.0001$, right hemisphere: $t(13) = 8.62$, $p < 0.0001$; Ventral - left hemisphere: $t(13) = 4.84$, $p = 0.0003$, right hemisphere: $t(13) = 7.45$, $p < 0.0001$; Medial - left hemisphere: $t(13) = 2.86$, $p = 0.013$, right hemisphere: $t(13) = 2.36$, $p = 0.034$).

276
277
278
279
280
281
282
283
284

4) Finally, I really liked the new figure 2, which anchors their findings compared to other prevalent atlases used in the field. However, I have 2 concerns regarding this new figure: (i) they use a much higher threshold here $t > 7.3$ compared to $t > 3.3$ in the other figures. I see no reason to change the thresholds across figures; So please update all figures to use the same threshold. (ii) Please show in additional panels the place-perception contrast from your study compared to the Glasser and Wang atlases, again using the same threshold.

We thank the reviewer for appreciating the group analysis shown in Figure 2.

285
286
287
288
289
290
291
292
293
294
295
296
297
298
299
300
301
302
303
304
305

Below is the suggested addition to the figure. As the reviewer hypothesized, the scene-perception localizer peaks on the ventral surface fell within the PHC1-2 ROIs (Wang et al 2014) and in the posterior portion of the PHC ROIs (Glasser 2016). Notably, these peaks are posterior to the peaks of group activation for the place-memory localizer. While we agree that this analysis is a valuable addition to the manuscript, we think that including it in the Main Text obscures the purpose of Figure 2, which is to demonstrate the location of the peaks of place-memory selective activation relative to these standard atlases. We have therefore included the group scene-perception area localizer results in the Extended data.

The reviewer also suggests changing the threshold for our exploratory group analysis to match the thresholds for the figures that show individual participant data. We respectfully disagree with this suggestion. This threshold was chosen to achieve an appropriate family-wise error correction value at the group level ($q = 0.00015$). Lowering the threshold would significantly increase the likelihood of showing false positive vertices, which is a major concern for fMRI studies that employ group analyses (Eklund et al. 2016). Notably, this is not a concern for Experiments 2-4, which employ single-subject ROI-based analyses. Additionally, the purpose of Figure 2 is to show the peak in activity of place-memory compared to the established anatomical and retinotopic atlases. Lowering the threshold would obscure these activation foci.

306
307
308
309
310
311
312
313

Extended data fig. 6. Location of scene perception activity in relation to known functional and anatomical landmarks. left. Unlike the place-memory areas, the scene-perception fall largely within retinotopic cortex. We conducted a group analysis of the scene-perception localizer (place > people memory recall; thresholded at vertex-wise $t > 7.3$, $p = 1e-6$) and compared the resulting activation to the most probable location of the cortical retinotopic maps using the atlas (overlaid) from Wang et al. 2014.

314 Right. Unlike the place-memory areas, the scene-perception areas fall in areas of cortex associated with
315 perception based on their location relative to a widely used anatomical parcellation (Ventral: PHC 1-3,
316 Medial: dorsal transitional visual area, Lateral: LO1/LO2, V3; Glasser et al. 2016).
317
318
319

Reviewers' Comments:

Reviewer #1:

Remarks to the Author:

The authors have done a rigorous revision and have addressed my remaining concerns. I appreciate that they added Extended Figure 7 as I believe that visualizing where the ROIs are for which the data is reported from is key for transparent reporting.